# The transcription factor hepatocyte nuclear factor 4A acts in the intestine to promote white adipose tissue energy storage

Romain Girard[1], Sarah Tremblay[1], Christophe Noll[2], Stéphanie St-Jean[1], Christine Jones[1], Yves Gélinas[3], Faïza Maloum-Rami[1], Nathalie Perreault [1], Mathieu Laplante [3,4], André C. Carpentier [2] & François Boudreau [1✉]

The transcription factor hepatocyte nuclear factor 4 A (HNF4A) controls the metabolic features of several endodermal epithelia. Both HNF4A and HNF4G are redundant in the intestine and it remains unclear whether HNF4A alone controls intestinal lipid metabolism. Here we show that intestinal HNF4A is not required for intestinal lipid metabolism per se, but unexpectedly influences whole-body energy expenditure in diet-induced obesity (DIO). Deletion of intestinal HNF4A caused mice to become DIO-resistant with a preference for fat as an energy substrate and energetic changes in association with white adipose tissue (WAT) beiging. Intestinal HNF4A is crucial for the fat-induced release of glucose-dependent insulinotropic polypeptide (GIP), while the reintroduction of a stabilized GIP analog rescues the DIO resistance phenotype of the mutant mice. Our study provides evidence that intestinal HNF4A plays a non-redundant role in whole-body lipid homeostasis and points to a non-cell-autonomous regulatory circuit for body-fat management.

[1] Department of Immunology and Cell Biology, Faculty of Medicine and Health Sciences, Université de Sherbrooke, Sherbrooke, QC, Canada. [2] Department of Medicine, Division of Endocrinology, Faculty of Medicine and Health Sciences, Université de Sherbrooke, Sherbrooke, QC, Canada. [3] Centre de recherche de l'Institut universitaire de cardiologie et de pneumologie de Québec, Faculty of Medicine, Université Laval, Québec, QC, Canada. [4] Centre de recherche sur le cancer de l'Université Laval, Université Laval, Québec, QC, Canada. ✉email: Francois.Boudreau@USherbrooke.ca

O besity represents a major worldwide epidemic problem, with more than 1.3 billion adults being considered overweight[1]. Being overweight is closely linked to metabolic disorders affecting several organs, including adipose tissue, pancreas, and liver. Diabetes is an important outcome associated with metabolic syndrome. Although there is active research to better define the mechanistic impact of these organs on the physiopathology of diabetes[2], relatively little attention has been given to the intestinal epithelium, the source of absorption of energy resources and their distribution to the organism.

As the first organ to sense and transit nutrients from the lumen into circulation, the digestive system also represents the largest endocrine organ, with the capacity to secrete more than 20 active hormones that influence metabolic function[3]. More specifically, incretins are gut-produced hormones with major functions in nutrition and metabolism[3]. GIP and glucagon-like peptide 1 (GLP-1) stimulate insulin secretion, whereas ghrelin restricts insulin secretion by targeting β cells of the pancreas. Because of the specific roles of incretins in glucose homeostasis, there has been a longstanding interest in exploring incretin-based therapies for type 2 diabetes (T2DM)[4,5].

Nuclear receptors belong to a superfamily of ligand-dependent transcription factors that play crucial roles in metabolic disorders[6]. Hepatocyte nuclear factor 4A (HNF4A; also designated maturity-onset diabetes of the young 1 (MODY1)) is a transcription factor of the nuclear receptor family that is expressed in multiple epithelial tissues of the liver, gut, pancreas, and kidney. Hepatic HNF4A is central to the regulation of glycolytic enzymes, glucose transport, and lipid metabolism[7,8]. HNF4A variants are associated with metabolic diseases including MODY1[9]. As opposed to its hepatic functions, intestinal HNF4A has redundant functions with its paralog HNF4G under normal physiological conditions. When HNF4A and HNF4G are simultaneously deleted in the intestinal epithelium, lethal intestinal defects are produced, in contrast to the minimal short-term phenotypic consequence of single-deleting either gene alone[10–13]. However, it is still unclear whether intestinal HNF4A alone is functionally involved in intestinal lipid metabolism, as it is observed in the liver.

To investigate the intestinal role of HNF4A in this context, we used a murine conditional deletion (HNF4A$^{\Delta IEC}$) previously shown to exhibit minor phenotypes without impacting nutrition, whole-body energy metabolism, and weight management under a normal diet[12,13]. This mouse model uses the VillinCre transgene to drive Cre expression exclusively in the intestinal epithelium since homozygosity for the unconditional null Hnf4a allele is embryonic lethal[14]. Here, we show that tissue-specific deletion of Hnf4a in the gut does not influence enterocyte lipid metabolism. However, we found that it produced resistance to diet-induced obesity (DIO) via non-cell-autonomous mechanisms involving GIP.

## Results

**HNF4A$^{\Delta IEC}$ mice are resistant to high-fat diet-induced obesity without apparent defects in intestinal lipid processing.** HNF4A$^{\Delta IEC}$ and control mice were fed a high-fat diet (HFD) for 12 weeks. We observed a significant reduction in HFD-induced weight gain in HNF4A$^{\Delta IEC}$ mutants, starting at 8 weeks post HFD feeding both in males (Fig. 1a) and females (Supplementary Fig. 1). Jejunal and liver tissues were isolated after euthanasia after 12 weeks of HFD exposure. Hnf4a transcript levels remained unchanged in the liver of HNF4A$^{\Delta IEC}$ mutants (Fig. 1b), while both Hnf4a transcripts (Fig. 1b) and protein (Fig. 1c) remained undetectable in the jejunum of HNF4A$^{\Delta IEC}$ mutants under HFD.

The expression of multiple intestinal lipid transporters was unchanged in HNF4A$^{\Delta IEC}$ jejunal samples compared to controls, as determined using RT-qPCR (Fig. 1d).

A transcriptomic analysis was performed to evaluate whether long-term HFD exposure influenced the whole jejunal transcriptome in HNF4A$^{\Delta IEC}$ mice. This analysis identified only 57 genes that were differentially expressed between HNF4A$^{\Delta IEC}$ and control mice (Fig. 1e). Gene ontology analysis identified very few biological processes, with regulation of peptide transport as the most significant functional annotation (FDR $< 1.5 \times 10^{-05}$) (Supplementary Fig. 2a). Gene set enrichment analysis (GSEA) confirmed that these genes were not significantly associated with lipid homeostasis (Fig. 1f), fat digestion and absorption (Fig. 1g), or lipid storage (Fig. 1h). These observations are consistent with the recent finding that both HNF4A and HNF4G deletions are required to impact lipid metabolism in the intestine[15]. Thus, HNF4A$^{\Delta IEC}$ mice harbored a phenotype of being resistance to DIO without a major impact on the intestinal transcriptome or lipid-metabolism signature.

**HNF4A$^{\Delta IEC}$ mice display better metabolic fitness during HFD exposure.** To better understand how HNF4A intestinal deletion reduced weight gain under HFD, we monitored the outcomes of feeding HNF4A$^{\Delta IEC}$ and control mice with HFD for a shorter period of 2 weeks. Food intake remained identical between HNF4A$^{\Delta IEC}$ and control mice during this period (Fig. 2a). Fecal triglyceride (TG) content similarly increased in both groups during the first days of HFD, allowing intestinal absorption to adapt to the newly introduced diet by 5 days, and became fully regularized after 14 days of HFD (Fig. 2b). Similarly, identical fat content was recovered from raw jejunum extracts (Fig. 2c), and similar fat content as measured using Oil Red O (ORO) staining was observed for both groups (Fig. 2d). HNF4A$^{\Delta IEC}$ mutants were significantly more resistant to weight gain as early as 4 days after commencement of HFD when ratios of the weights were plotted (Fig. 2e) as opposed to individual body weights (Supplementary Fig. 3). This observation was consistent with a significant 28% decrease in total body fat percentage in HNF4A$^{\Delta IEC}$ mutants fed an HFD for 2 weeks (Fig. 2f). We then investigated visceral adipose storage in these mice, given its preponderant influence on metabolic disease[16]. Histological analysis of epididymal WAT (eWAT) (Fig. 2g), which is considered to be a part of visceral fat in rodents, revealed a significantly higher proportion of smaller adipocytes in HNF4A$^{\Delta IEC}$ mutants (Fig. 2h). We then performed an insulin tolerance test (ITT) to define the impact of intestinal HNF4A loss on systemic insulin sensitivity. We found that HNF4A$^{\Delta IEC}$ mutants did not show differences in fasting blood glucose levels prior to insulin injection (Fig. 2i). However, intraperitoneal insulin reduced blood glucose in HNF4A$^{\Delta IEC}$ mutants below that of controls (15 min: 6.0 mM for controls, 2.6 mM for mutants, $P < 0.01$; 30 min: 5.6 mM for controls, 3.0 mM for mutants, $P < 0.05$) during the first 30 min of challenge, indicative of improved insulin sensitivity in HNF4A$^{\Delta IEC}$ mice after 2 weeks of HFD (Fig. 2i). Serum resistin levels, related to obesity and insulin resistance in mice[17], were also decreased in HNF4A$^{\Delta IEC}$ mutants (Fig. 2j). Obesity-related steatosis has been linked metabolically to insulin resistance and diabetes[18]. Immunolabeling of hepatic cytoplasmic lipid droplets (LDs) against the adipose differentiation-related protein (ADFP) showed similar size ranges of LDs, indicating no effect on hepatic fat deposition after 2 weeks of HFD (Fig. 2k). However, prolonged exposure to HFD for 12 weeks strengthened hepatic fat accumulation, particularly in control mice that presented with extended areas of oversized LDs, which were less common in mutants (45.1% for controls, 6.5% for mutants of the hepatic section area, $P < 0.05$)

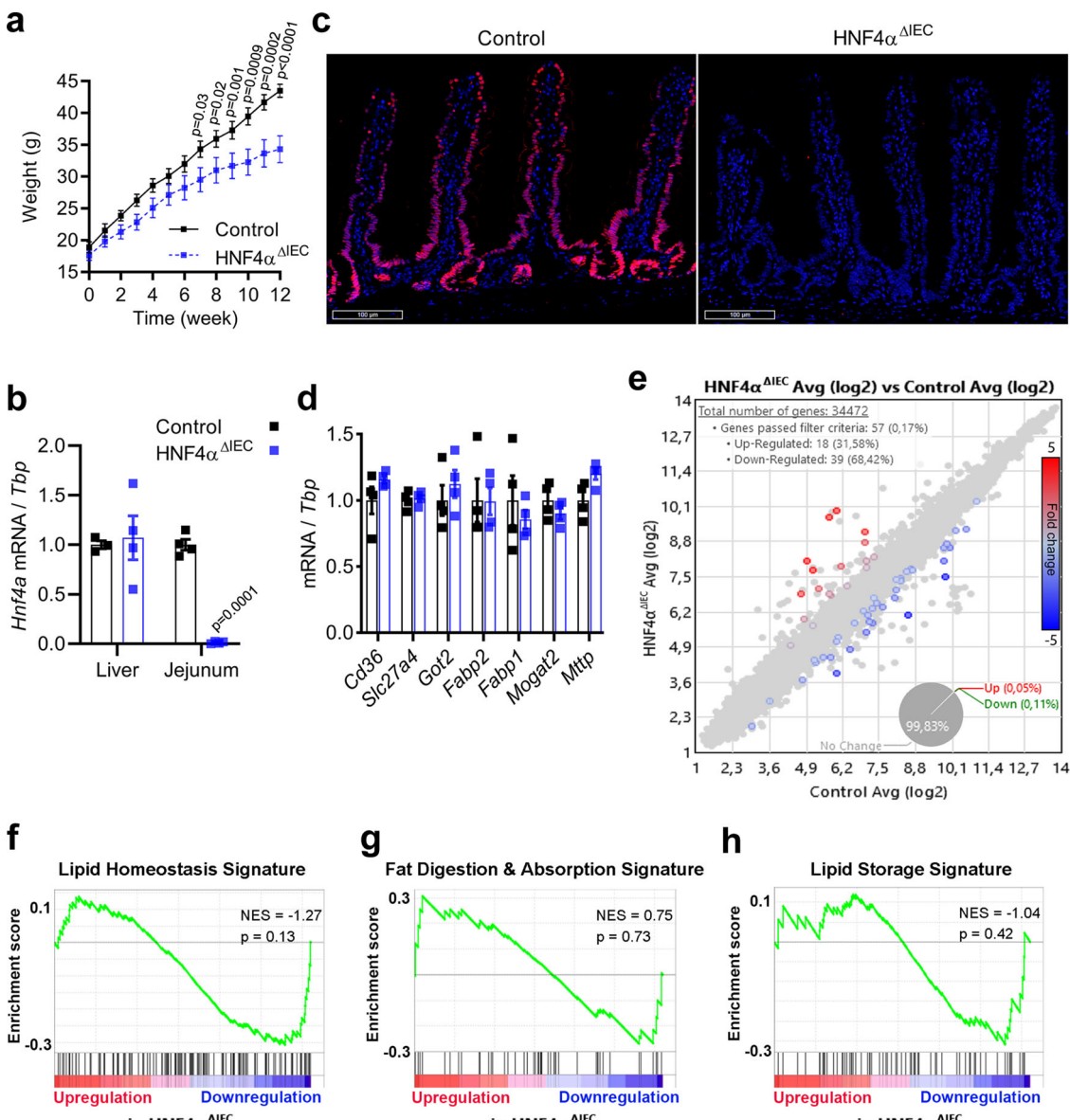

**Fig. 1 Mice lacking intestinal HNF4A are resistant to HFD-induced obesity without significant alteration in expression of genes related to lipid metabolism. a** Weights of weaned mice fed a HFD during 12 weeks for control (black squares) and HNF4A$^{\Delta IEC}$ mutant (blue squares) mice ($n = 10$). Statistical comparisons were performed using two-way ANOVA followed by uncorrected Fisher's LSD test. **b** Quantification of *Hnf4a* transcripts relative to *Tbp* measured in the liver and jejunum of control (black squares) and HNF4A$^{\Delta IEC}$ mutant (blue squares) mice fed a HFD ($n = 4$). Statistical comparisons were performed using two-tailed Mann–Whitney test. **c** Representative jejunum sections stained with DAPI and immunolabeled against HNF4A in control and HNF4A$^{\Delta IEC}$ mutant mice fed a HFD ($n = 3$). **d** Quantification of lipid transporter gene transcripts relative to *Tbp* measured in the jejunum of control (black squares) and HNF4A$^{\Delta IEC}$ mutant (blue squares) mice fed a HFD ($n = 4$). Statistical comparisons were performed using the Mann–Whitney $U$ test. **e** Scatter plot of gene microarray data from the jejuna of control and HNF4A$^{\Delta IEC}$ mutant mice fed a HFD for 12 weeks ($n = 3$). The colored dots indicate transcripts significantly upregulated (red) or downregulated (blue) in HNF4A$^{\Delta IEC}$ mutant relative to control mice according to filter criteria (fold change: >1 or <−1; FDR $P < 0.10$). GSEA of microarray data from the jejuna of control and HNF4A$^{\Delta IEC}$ mutant mice fed a HFD for 12 weeks showing no significant changes in expression of genes related to **f**, lipid homeostasis; **g**, fat digestion and absorption; and **h**, lipid storage. NES (normalized enrichment score) and nominal *p* value are depicted. Data are presented as mean values ± SEM. Source data are provided as a Source Data file.

(Fig. 2l). Altogether, these data show that HNF4A$^{\Delta IEC}$ mutants maintain dietary and intestinal uptake of TGs and fatty acids, but are less susceptible to DIO metabolic dysfunctions than control mice.

**HNF4A$^{\Delta IEC}$ mice display increased WAT metabolism, fat oxidation, and energy expenditure during HFD exposure**. Early changes observed in the eWAT histology of HNF4A$^{\Delta IEC}$ mutants under HFD led us to investigate changes in gene expression in

this tissue using qRT-PCR to measure gene transcripts associated with various metabolic lipid functions. Several transcripts involved in adipocyte insulin sensitivity, fatty-acid catabolism, mitochondrial biogenesis, and thermogenesis were upregulated in the eWAT of HNF4A$^{\Delta IEC}$ mutant mice (Fig. 3a). *Ucp1* and *Prdm16* were induced 14.5-fold ($P < 0.05$) and 1.5-fold ($P < 0.01$), respectively (Fig. 3b). Interestingly, UCP1, which is specific for BAT and beige adipose cells, is activated by long-chain fatty acids and acts as a mitochondrial proton carrier to generate heat instead of ATP[19].

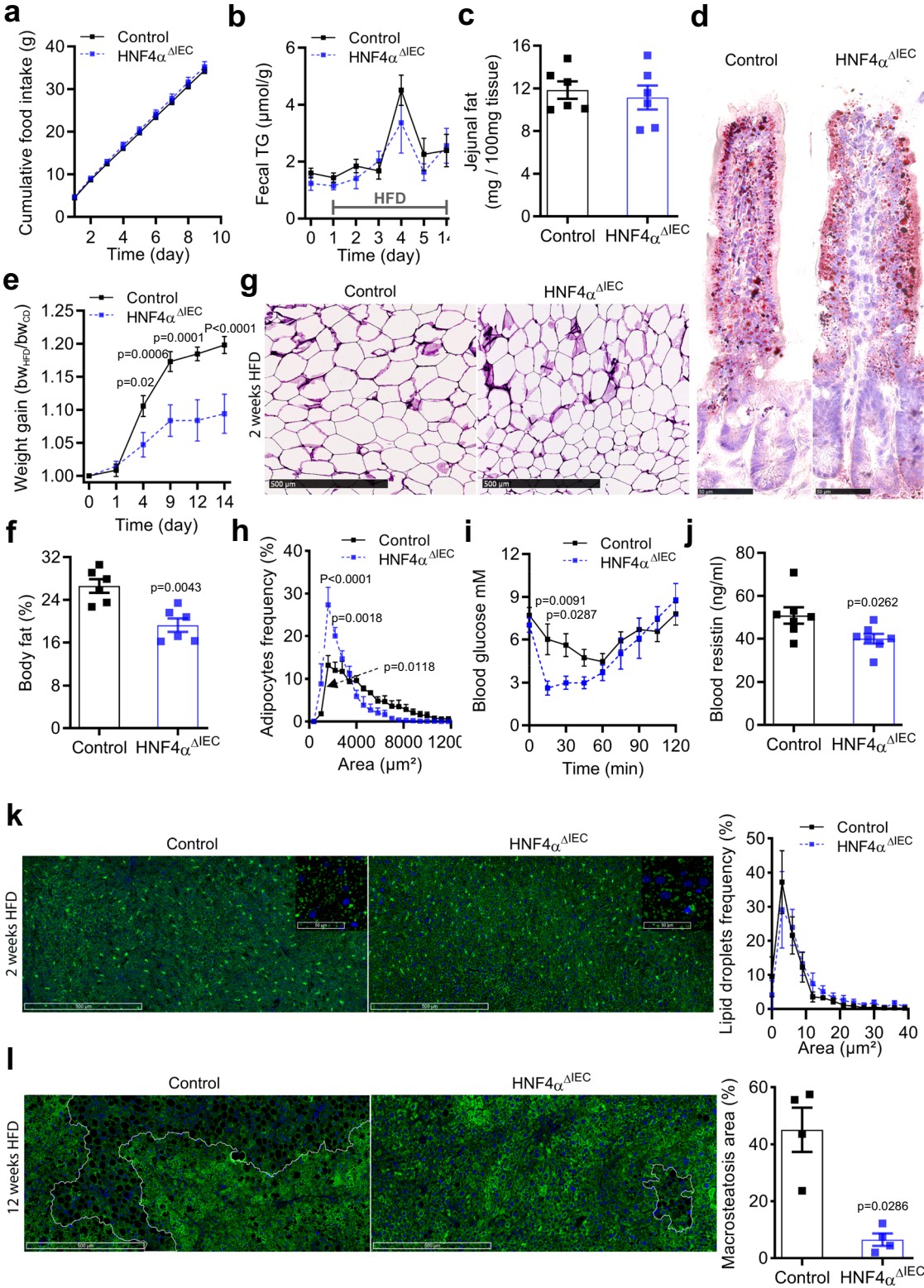

HNF4A$^{\Delta IEC}$ and control mice fed a HFD were then subjected to 18 fluorodeoxyglucose (FDG) positron emission tomography (PET)/computed tomography (CT) imaging to visualize metabolically active adipose tissues. BAT FDG uptake remained unchanged between HNF4A$^{\Delta IEC}$ and control mice (Fig. 4a). While inguinal WAT (iWAT) remained similar under these conditions, eWAT FDG uptake was significantly enhanced in HNF4A$^{\Delta IEC}$ mutant mice (2.57-fold; $P < 0.01$) (Fig. 4a). When

relative WAT FDG was reported to BAT FDG fractional uptake, a significant and progressive decrease from BAT toward iWAT and eWAT was observed in control mice (Fig. 4b), as previously described[20]. In contrast, FDG avidity from both iWAT and eWAT did not differ from BAT in HNF4A$^{\Delta IEC}$ mutants (Fig. 4b), suggesting an increase in the overall WAT metabolic activities that meet the BAT normal activity. Collectively, WAT expression data and metabolic activities indicated WAT beiging of

**Fig. 2 Loss of intestinal epithelial HNF4A improves metabolic fitness during both short and long-term HFD exposure. a** Cumulative food intake monitored in metabolic cages for 9 consecutive days for both adult control (black squares) and HNF4A$^{\Delta IEC}$ mutant (blue squares) mice fed a HFD ($n = 5$). Statistical comparisons were performed using two-way ANOVA followed by uncorrected Fisher's LSD test. **b** Triglyceride levels in fecal lipid extracts measured using colorimetry in control (black squares) and HNF4A$^{\Delta IEC}$ mutant (blue squares) mice fed a HFD ($n = 6$). Statistical comparisons were performed using two-way ANOVA followed by uncorrected Fisher's LSD test. **c** Raw fat mass extracted from jejunum tissues of control (black squares) and HNF4A$^{\Delta IEC}$ mutant (blue squares) mice fed a HFD for 2 weeks ($n = 6$). Statistical comparisons were performed using two-tailed Mann–Whitney test. **d** Representative jejunum villi stained with Oil Red O (ORO) from control and HNF4A$^{\Delta IEC}$ mutant mice fed a HFD for 2 weeks ($n = 3$). **e** Weight gain ratios of control (black squares) and HNF4A$^{\Delta IEC}$ mutant (blue squares) mice fed a HFD for 2 weeks ($n = 6$). Statistical comparisons were performed using two-way ANOVA followed by uncorrected Fisher's LSD test. **f** Percentage of total body fat of control (black squares) and HNF4A$^{\Delta IEC}$ mutant (blue squares) mice fed a HFD for 2 weeks, from DXA analysis ($n = 6$). Statistical comparisons were performed using two-tailed Mann–Whitney test. eWAT histological sections (**g**) and histomorphometric analysis (**h**) from controls (black squares; $n = 4$) and HNF4A$^{\Delta IEC}$ mutant (blue squares; $n = 3$) mice fed a HFD for 2 weeks (three sections analyzed per sample). Statistical comparisons were performed using two-way ANOVA followed by uncorrected Fisher's LSD test. **i** Insulin tolerance test (ITT) performed on control (black squares; $n = 9$) and HNF4A$^{\Delta IEC}$ mutant (blue squares; $n = 5$) mice fed a HFD for 2 weeks. Statistical comparisons were performed using two-way ANOVA followed by uncorrected Fisher's LSD test. **j** Blood resistin levels measured using ELISA in control (black squares) and HNF4A$^{\Delta IEC}$ mutant (blue squares) mice fed a HFD for 2 weeks ($n = 7$). Statistical comparisons were performed using two-tailed Mann–Whitney test. **k–l** Representative hepatic sections revealing lipid droplets (LDs) immunostained for ADFP (green) and counterstained with DAPI. **k** Homogeneity of hepatic LDs and size analysis for control (black squares) and HNF4A$^{\Delta IEC}$ mutant (blue squares) mice fed a HFD for 2 weeks ($n = 4$). Statistical comparisons were performed using two-way ANOVA followed by uncorrected Fisher's LSD test. **l** Heterogeneity of hepatic LD size and occurrence in areas of macrosteatosis (white circles) calculated for control (black squares) and HNF4A$^{\Delta IEC}$ mutant (blue squares) mice fed a HFD for 12 weeks ($n = 4$). Statistical comparisons were performed using two-tailed Mann–Whitney test. Data are presented as mean values ± SEM. Source data are provided as a Source Data file.

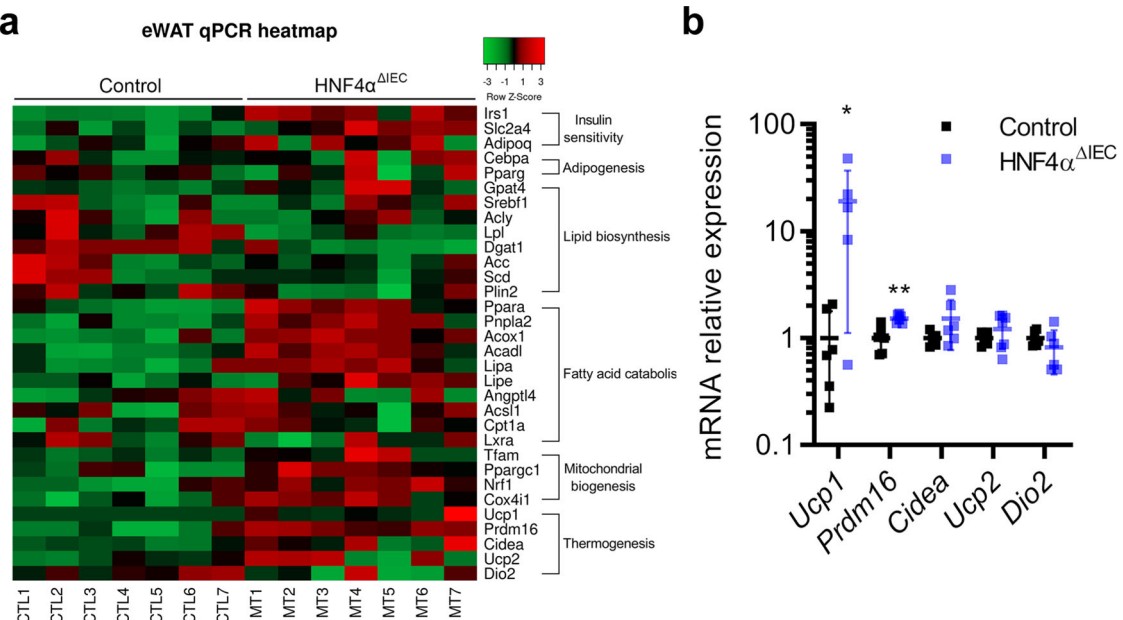

**Fig. 3 WAT metabolic gene signature in HNF4A$^{\Delta IEC}$ mutants under HFD. a** Heatmap from RT-qPCR quantification of eWAT genes from control and HNF4A$^{\Delta IEC}$ mutant mice fed a HFD for 2 weeks ($n = 7$). The green to red color scale indicates intensity of negative to positive row $Z$-Score. **b** RT-qPCR quantification of thermogenesis-related genes from control (black squares) and HNF4A$^{\Delta IEC}$ mutant (blue squares) mice fed a HFD for 2 weeks ($n = 7$). Statistical comparisons were performed using two-tailed Mann–Whitney test. Data are presented as mean values ± SEM. Source data are provided as a Source Data file.

HNF4A$^{\Delta IEC}$ mutants under HFD. There is evidence supporting that WAT beiging and increased thermogenesis can lead to increased energy expenditure (EE)[21,22]. The effect of deleting intestinal epithelial HNF4A on the EE of mice fed a HFD was assessed using indirect calorimetry. Oxygen consumption was elevated in the HNF4A$^{\Delta IEC}$ mutants as monitored for 3 days ($P < 0.05$; Fig. 4c). When the data were specifically analyzed into light-dark circadian cycles, oxygen consumption was mainly enhanced during the light phase (Fig. 4d), indicating higher oxidative rates even during the resting period of mutant mice. The respiratory exchange ratio (RER) remained significantly below the control baseline during the active dark phase of the HNF4A$^{\Delta IEC}$ mutants (Fig. 4e). A lower RER is indicative of a metabolic switch to the preferential use of fat as an energy source[23]. The fat oxidation rate was calculated. As expected, control mice harbored a cyclic use of fat that is usually favored during resting and prolonged fasting periods but restrained during the use of carbohydrates during their active periods, with a significant difference between light and dark fat oxidation rates (Fig. 4f; $P < 0.05$)[24]. HNF4A$^{\Delta IEC}$ mutants lost this circadian rhythm ($P = 0.43$) and maintained higher fat oxidation rates, especially during the dark active phase (Fig. 4f). As a result, EE increased in mutant mice (Fig. 4g) and body temperature increased significantly under these conditions, potentially induced by elevated thermogenesis (Fig. 4h). Taken together, these data support a functional role for intestinal HNF4A in

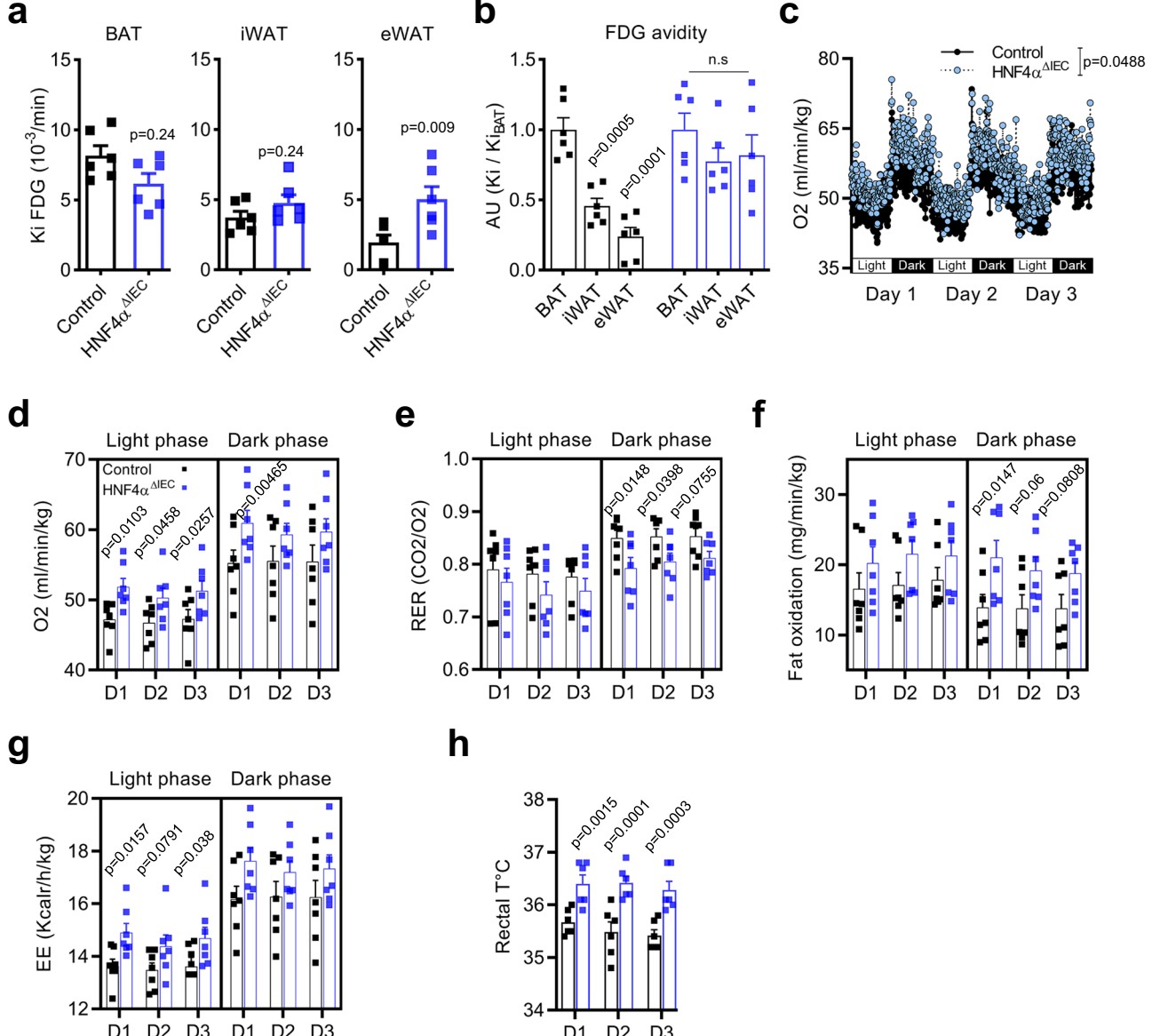

**Fig. 4 Loss of intestinal epithelial HNF4A impacts WAT metabolism, fat oxidation, and energy expenditure in mice fed a HFD. a** PET estimates of metabolic rates based on FDG uptake (Ki) by BAT, iWAT, and eWAT from control (black squares) and HNF4A$^{\Delta IEC}$ mutant (blue squares) mice fed a HFD for 2 weeks ($n = 6$ biologically independent animals). Statistical comparisons were performed using two-tailed Mann–Whitney test. **b** Relative WAT FDG were reported to BAT FDG fractional uptake ($n = 6$) Statistical comparisons were performed using two-way ANOVA followed by uncorrected Fisher's LSD test. **c** Oxygen consumption from indirect calorimetry of control (black circles) and HNF4A$^{\Delta IEC}$ mice (blue circles) fed a HFD for 2 weeks ($n = 7$) Statistical comparisons were performed using two-way ANOVA test. Averages of oxygen consumption (**d**), respiratory exchange ratio RER (**e**) fat oxidation (**f**), and energy expenditure (EE) (**g**) per daily phases for the last 3 days of HFD for control (black squares) and HNF4A$^{\Delta IEC}$ mice (blue squares) ($n = 7$) Statistical comparisons were all performed using two-way ANOVA followed by uncorrected Fisher's LSD test. **h** Rectal temperatures of control (black squares) and HNF4A$^{\Delta IEC}$ mutant (blue squares) mice during the last 3 days of HFD ($n = 6$). Statistical comparisons were performed using two-way ANOVA followed by uncorrected Fisher's LSD test. Data are presented as mean values ± SEM. Source data are provided as a Source Data file.

reducing WAT metabolic activity, fat oxidation, and EE under a HFD.

**DIO resistance phenotype of HNF4A$^{\Delta IEC}$ mutants is rescued with GIP analog exposure.** Our data support a role for intestinal HNF4A in WAT metabolism via non-cell-autonomous action involving a signaling molecule derived from the intestinal epithelium. Gene ontology identified multiple potential HNF4A targets (Supplementary Fig. 2b). Among them, *GIP* is transcriptionally regulated by HNF4A[13] and we observed a reduction

in *GIP* transcripts in the jejunum of HNF4A$^{\Delta IEC}$ mutants fed a HFD (2.7-fold, $P < 0.05$) (Fig. 5a).

Since GIP is secreted at a higher rate in obese mice[25] and its release is sensitive to fat ingestion[26], we assessed GIP response to an olive-oil bolus in HNF4A$^{\Delta IEC}$ mutants fed a HFD. Control mice displayed a rapid and marked increase in GIP release during this oral fat tolerance test (OFTT). HNF4A$^{\Delta IEC}$ mutant mice displayed blunted GIP release under these conditions (Fig. 5b). Because disruption of GIP signaling has previously been linked to DIO resistance in mice[27,28], we took advantage of an enzymatically stable analog of GIP, (D-Ala$^2$)GIP(Lys$^{37}$PAL), with

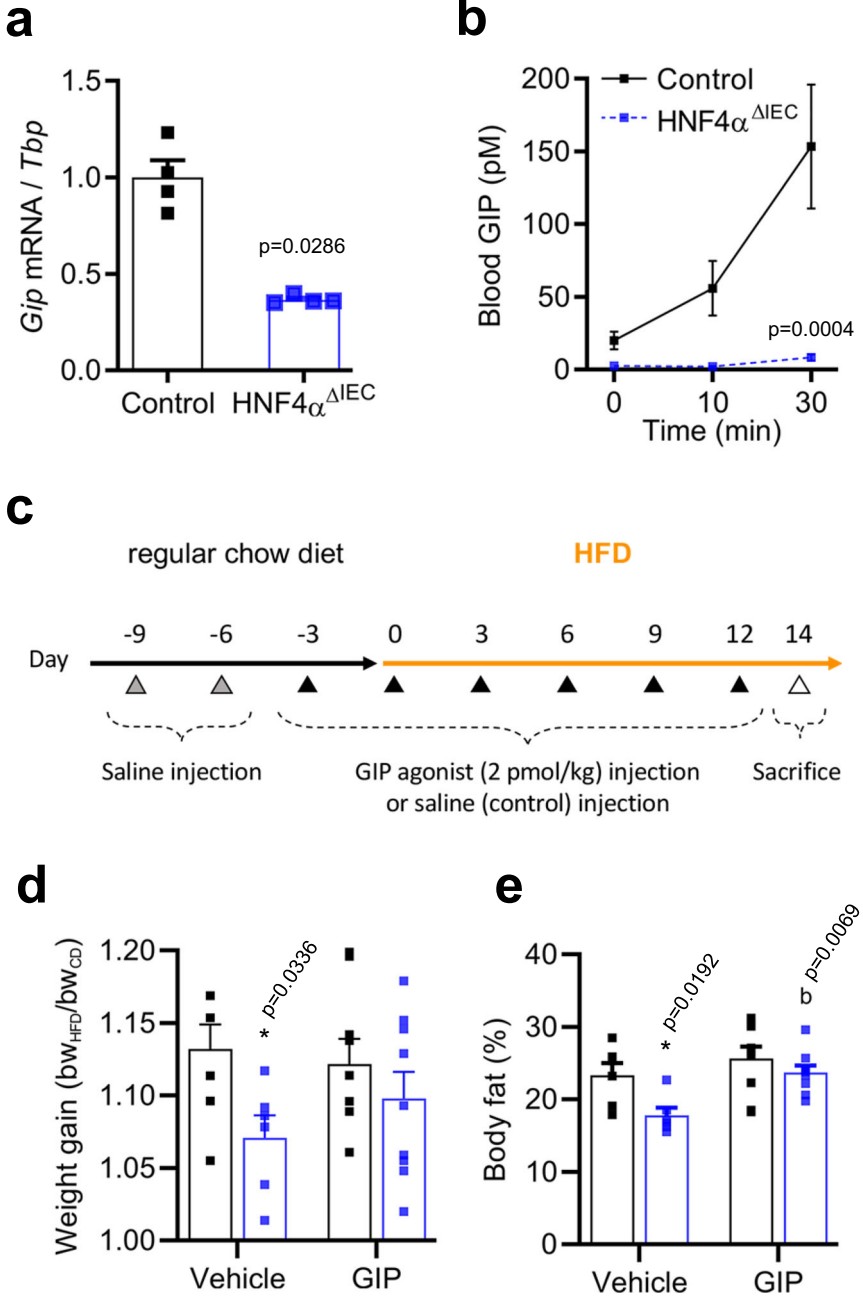

**Fig. 5 GIP analog exposure rescues the DIO resistance phenotype of HNF4A$^{\Delta IEC}$ mutants. a** *Gip* transcript expression relative to *Tbp* in the jejunum of HFD fed control (black squares) and HNF4A$^{\Delta IEC}$ mutant (blue squares) mice ($n = 4$). Statistical comparisons were performed using two-tailed Mann–Whitney test. **b** Circulating GIP levels were determined using ELISA after OFTT in HFD fed control (black squares; $n = 8$) and HNF4A$^{\Delta IEC}$ mutant (blue squares; $n = 5$) mice. Statistical comparisons were performed using the two-way ANOVA followed by uncorrected Fisher's LSD test. **c** Protocol timeline of HFD feeding and (D-Ala$^2$)GIP[Lys$^{37}$PAL] injections in both control and HNF4A$^{\Delta IEC}$ mutant mice. Weight gain (**d**) and body fat amount (**e**) of control (black squares) and HNF4A$^{\Delta IEC}$ mutant mice (blue squares) injected with saline ($n = 7$ for controls and $n = 6$ for mutants) or (D-Ala$^2$)GIP[Lys$^{37}$PAL] ($n = 9$ for both controls and mutants) at 2 pmol/kg and fed a HFD for 2 weeks. Statistical comparisons were performed using the two-way ANOVA followed by uncorrected Fisher's LSD test. Asterisk indicates comparison versus control and mutant mice from the saline group, and b indicates comparison versus mutant mice from the saline and the GIP analog group. Data are presented as mean values ± SEM. Source data are provided as a Source Data file.

confirmed bioactivity and antidiabetic effects in mice fed an HFD[29]. HNF4A$^{\Delta IEC}$ mutant and control mice were injected for 24 days with saline or (D-Ala$^2$)GIP(Lys$^{37}$PAL) at a dose of 2 pmol/kg (Fig. 5c), estimated to be comparable to physiological levels and to what has been done before in rodents[30]. Weight gains of HNF4α$^{\Delta IEC}$ mutant and control mice treated with (D-Ala$^2$)GIP (Lys$^{37}$PAL) were statistically indistinguishable under

these conditions, whereas HNF4A$^{\Delta IEC}$ mutants displayed a significant reduction in weight gain when exposed to saline (5%; $P < 0.05$) (Fig. 5d). Coincidently, the body fat composition of HNF4A$^{\Delta IEC}$ mutant and control mice remained similar under (D-Ala$^2$)GIP(Lys$^{37}$PAL) treatment, as opposed to a 24% reduction in total body fat content in HNF4A$^{\Delta IEC}$ mutants injected with saline (Fig. 5e), as observed above (Fig. 2f). These data

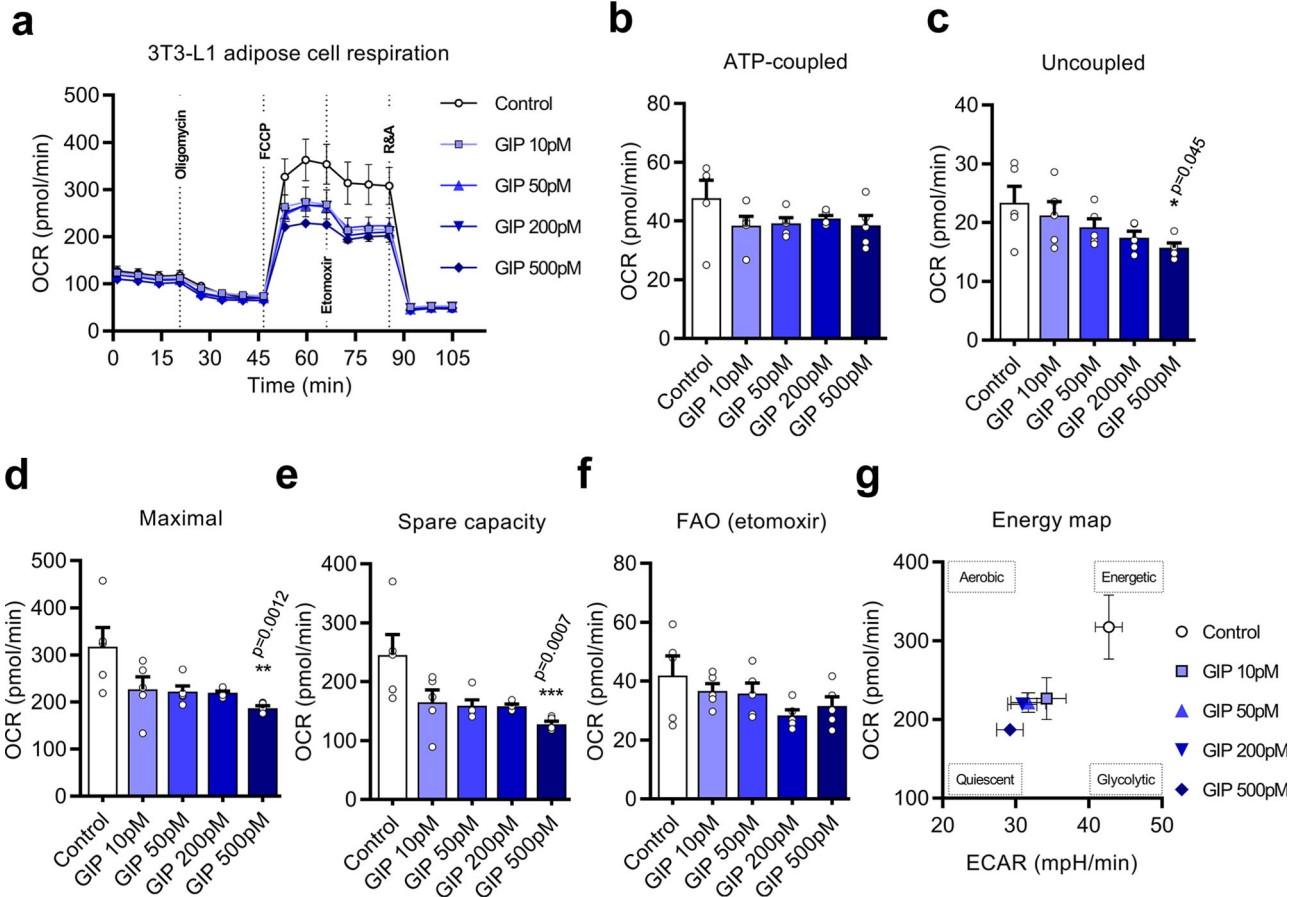

**Fig. 6 GIP analog exposure reduces metabolic rates of differentiated 3T3-L1 white adipocytes. a** Seahorse oxygen consumption rate (OCR) data acquisition of differentiated 3T3-L1 cells exposed to gradient doses of (D-Ala²)GIP[Lys³⁷PAL] (control: white circles, 10 pM: lavender blue squares, 50 pM: light blue triangles, 200 pM: blue inverted triangles, 500 pM: dark blue diamonds) ($n = 5$ biological independent samples over two independent experiments). ATP-coupled (**b**), uncoupled (**c**), maximal (**d**), spare capacity (**e**), and dedicated fatty-acid oxidation (FAO) (**f**) respiration parameters of differentiated 3T3-L1 cells exposed to (D-Ala²)GIP[Lys³⁷PAL] during mitochondrial stress tests ($n = 5$ biological independent samples over two independent experiments, control: white columns, 10 pM: lavender blue columns, 50 pM: light blue columns, 200 pM: blue columns, 500 pM: dark blue columns). **g** Incidence of (D-Ala²)GIP[Lys³⁷PAL] exposures upon metabolic performances of 3T3-L1 differentiated adipocytes (control: white circles, 10 pM: lavender blue squares, 50 pM: light blue triangles, 200 pM: blue inverted triangles, 500 pM: dark blue diamonds). Scatter plot of OCR against extracellular acidification rate (ECAR) as determined during the maximal mitochondrial respiration phase ($n = 5$ biological independent samples over two independent experiments). Statistical comparisons for all panels were performed using Kruskal–Wallis followed by Dunn's test. Data are presented as mean values ± SEM. Source data are provided as a Source Data file.

indicate that stabilization of GIP circulating levels in HNF4A^ΔIEC mutants eliminates their resistance to fat mass gain when fed a HFD.

**GIP analog exposure reduces metabolic rates of white adipocytes.** We then investigated whether GIP signaling directly affects adipose metabolism. Murine 3T3-L1 adipocytes are widely used as a WAT model and have been found to be typical of white adipocytes[31]. In addition, these cells have been shown to be GIP sensitive, with a positive impact on glucose uptake and adipocyte development[32]. The 3T3-L1 cells were differentiated in culture and exposed to increasing concentrations of (D-Ala²)GIP (Lys³⁷PAL) to monitor the oxygen consumption rate (OCR), supplementing with chemical effectors that challenge mitochondrial function (Fig. 6a). While ATP-coupled respiration remained statistically unchanged regardless of (D-Ala²)GIP[Lys³⁷PAL] supplementation (Fig. 6b), the uncoupled respiration of differentiated adipocytes was significantly lower at 500 pM (D-Ala²) GIP(Lys³⁷PAL) (1.5-fold, $P < 0.05$) (Fig. 6c). Uncoupled respiration is a core feature of brown and beige fat that generates heat at the expense of high energy cost[19,33]. Maximal mitochondrial

respiration induced by carbonyl cyanide-4 (trifluoromethoxy) phenylhydrazone (FCCP) was reduced in differentiated adipocytes upon exposure to increasing concentrations of (D-Ala²) GIP(Lys³⁷PAL), notably at 500 pM (1.7-fold, $P < 0.01$) (Fig. 6d). Similarly, spare respiratory capacity was also reduced by (D-Ala²) GIP(Lys³⁷PAL) at 500 pM (1.9-fold, $P < 0.001$) (Fig. 6e), indicating a reduced ability of adipocytes to meet an increased energy demand under these conditions. (D-Ala²)GIP(Lys³⁷PAL) supplementation of differentiated 3T3-L1 cells did not affect OCR driven by fatty-acid oxidation (FAO), as determined by the addition of etomoxir, a carnitine palmitoyltransferase-1 inhibitor (Fig. 6f). Finally, (D-Ala²)GIP(Lys³⁷PAL) was able to switch the bioenergetic profile of adipocytes from energetic to quiescent, as measured by the OCR/extracellular acidification rate (ECAR) ratio (Fig. 6g).

To further investigate the effect of (D-Ala²)GIP(Lys³⁷PAL) on WAT metabolism, mice were exposed to the same protocol described above (Fig. 5c). A transcriptomic analysis was performed on both iWAT and eWAT from mice injected or not injected with the agonist. This analysis identified 562 genes for iWAT (Fig. 7a) and 75 genes for eWAT (Fig. 7b) that were

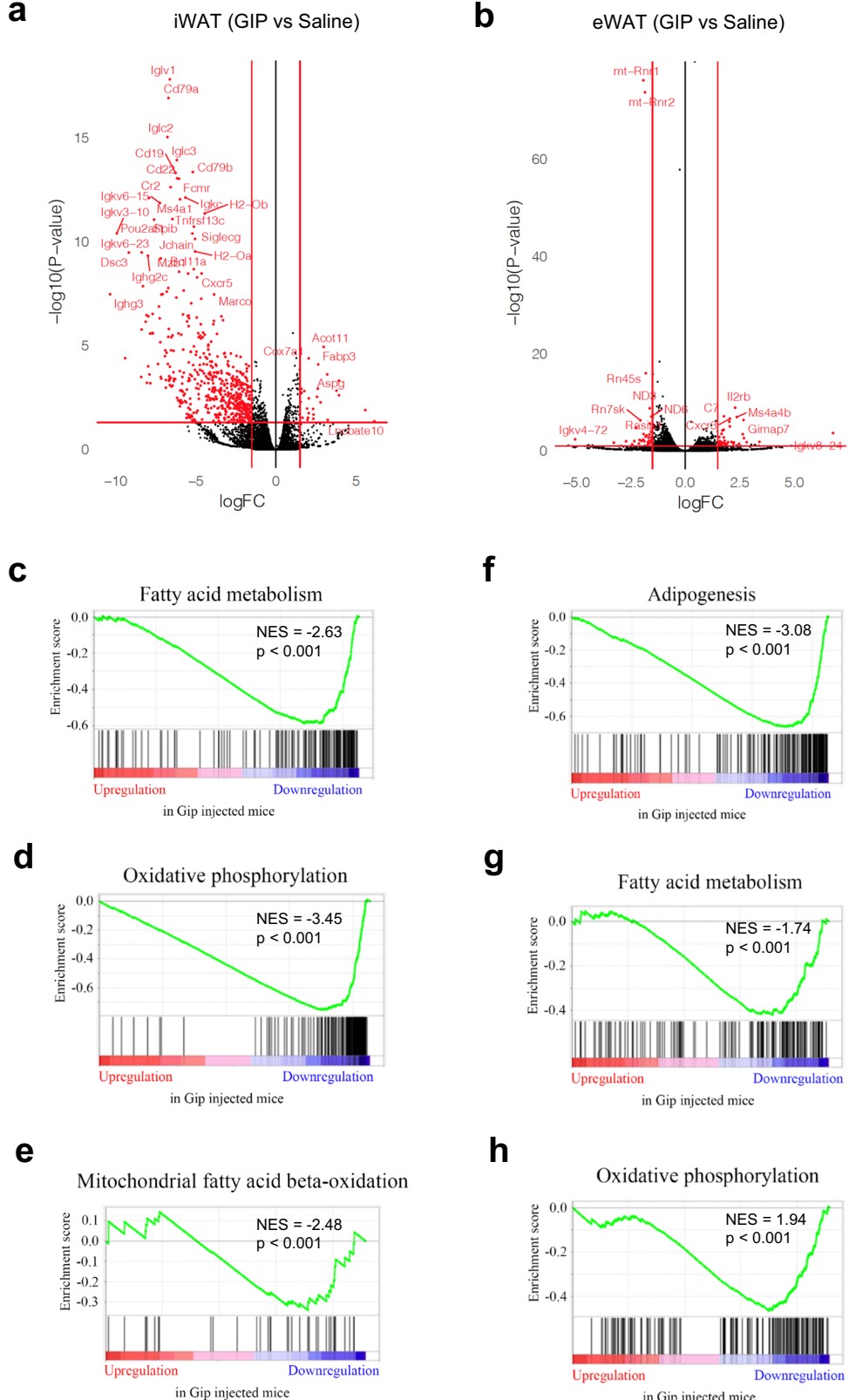

**Fig. 7 GIP analog exposure influences WAT transcriptome in mice fed a HFD.** Volcano plots of differential gene expression obtained from RNA-seq of iWAT (**a**) and eWAT (**b**) after GIP analog exposure. iWAT RNA-seq data were analyzed using GSEA to evaluate changes in fatty-acid metabolism (**c**), oxidative phosphorylation (**d**), mitochondrial fatty-acid beta-oxidation (**e**) and adipogenesis (**f**). eWAT RNA-seq data were also analyzed for changes in fatty-acid metabolism (**g**) and oxidative phosphorylation (**h**). NES (normalized enrichment score) and nominal *p* value are depicted.

differentially expressed (FDR < 0.05) between control and (D-Ala$^2$)GIP(Lys$^{37}$PAL)-injected mice. GSEA showed a significant reduction in fatty-acid metabolism (Fig. 7c), oxidative phosphorylation (Fig. 7d), mitochondrial beta-oxidation (Fig. 7e), and adipogenesis (Fig. 7f) of iWAT after agonist treatment. Similar results were also observed in eWAT, with significant reductions in fatty-acid metabolism (Fig. 7g) and oxidative phosphorylation (Fig. 7h).

Taken together, our data point to a mechanism for the GIP analog negatively affecting the metabolic rate of white adipocytes, resulting in a reduction in EE and an increase in fat energy storage in the WAT of mice fed a HFD.

## Discussion

HNF4A, as observed for many nuclear receptors, plays a central role in regulating metabolic features. For instance, HNF4A regulates insulin signaling in the pancreas and carbohydrate and fatty-acid metabolism in the liver. There is clinical interest in positively targeting HNF4A in drug metabolism and fatty liver diseases[34,35]. Our findings point to a nonessential role of HNF4A in regulating fatty-acid transport and metabolism in the intestinal epithelium. Thus, it is likely that lipid handling and metabolism in enterocytes are covered by redundant roles of HNF4 orthologs[11,15]. However, our work identified a pivotal non-cell-autonomous function for gut HNF4A by itself in controlling fat usage under DIO. GIP is exclusively produced by the small intestine, and we recently showed that HNF4A acts as a positive regulator of *Gip* transcription[13]. We also found a modest reduction in circulating GIP levels after an oral glucose tolerance test, with no effect on glucose homeostasis in mice deficient for HNF4A in the intestine[13]. These observations contrast with our observation of GIP secreted levels being drastically reduced in HNF4A$^{\Delta IEC}$ mutant mice after lipid exposure. In addition to its transcriptional role on *Gip*, it is most likely that HNF4A plays a complementary role in GIP secretion after exposure to a lipid bolus; however, the exact nature of these mechanisms awaits further elucidation.

Unlike the liver, our data indicate a beneficial effect of inhibiting HNF4A in a gut-specific manner in minimizing the impact of DIO. These data are consistent with the lean phenotype of MODY1 patients, a feature that distinguishes itself from T2DM patients with obesity[36]. Although currently, there has been no direct functional association reported between HNF4A and GIP expression in the context of these diseases, some case reports have suggested possible relationships between GIP altered circulating levels in MODY1 patients. One described the discovery of *RFX6* variants enriched in a MODY1 cohort that were associated with lower fasting and stimulated levels of GIP[37]. Another report described a young patient bearing a mutation in *HNF4A* (MODY1) with an impaired incretin response that includes GIP[38]. Determining whether such a regulatory circuit between HNF4A and GIP is of functional importance in normal and diseased individuals with obesity requires further investigation.

GIP signaling has previously been shown to contribute to DIO in multiple mouse models. *Gipr*$^{-/-}$ mice are resistant to DIO with high similarities to HNF4A$^{\Delta IEC}$ mutants[27], while deletion of GIPR specifically in WAT results in similar DIO resistance features including reduced body weight, insulin resistance and hepatic steatosis[39]. Our data demonstrate that in the absence of gut HNF4A, mice fed a HFD displayed an increase in overall WAT metabolic activities at comparable levels to BAT activity. In addition, the injection of a GIP analog potently influenced the WAT transcriptome with modifications associated with a reduction in fatty-acid metabolism, oxidative phosphorylation, and mitochondrial beta-oxidation. Taken together, these observations support the natural positive adiposity potential of GIP in WAT. In accordance with this, *Gipr* expression has been reported to be more important in WAT than in BAT[40]. Whether GIP signaling preferentially inhibits the metabolic activity of WAT based on the availability of GIPR is possible in this context.

There is currently a great deal of controversy over the physiopathological impacts of pharmacologically targeting GIP in the context of weight management, as well as for its therapeutic potential during diabetes, metabolic syndrome, and obesity[41]. One study that used the same GIP agonist reported a lack of any effect on total body fat composition when administered to mice that were already obese[29]. More recently, optimized GIP analogs were able to modestly reduce body weight gain in steady-state obese mice; this effect was strictly dependent on food intake changes[42]. One important difference with our study is that our mouse model is severely impaired in GIP secretion upon exposure to dietary fat. Other data consistent with our observations support the anti-obesity effects of GIPR antagonists, at least in mice[43]. To add to this controversy, there is a great deal of divergence in the literature regarding the physiological versus supraphysiological doses for GIP agonists. In terms of animal physiology, the basal circulating level of GIP ranges between 10 and 40 pM and can reach 100–500 pM after lipid ingestion. The observed antidiabetic effects of some GIP agonists have been observed at doses up to a thousand times higher than the normal physiological circulating levels for GIP. We administered a GIP analog at close to the normal physiological level and thus our data must be interpreted in this context.

One important limitation of our study was its focus on male subjects. Although we observed that female HNF4A$^{\Delta IEC}$ mice displayed similar resistance to DIO, this observation does not warrant the conclusion that the same mechanism operates in both sexes, as recently noted[44]. This aspect deserves further consideration, especially in the design of clinical studies to follow up on our findings.

In conclusion, our results indicate a previously unsuspected role of HNF4A in positively contributing to adipogenesis via non-cell-autonomous mechanisms (Fig. 8). This insulinotropic effect of GIP is diminished in human patients, while administration of dual GLP1R/GIPR agonists showed beneficial effects on glycemic control and body weight in these patients. Whether GIP represents a gut-derived molecule with WAT beiging properties is a compelling prospect. Although our results support a negative effect of GIP on adipocyte mitochondrial respiration and energy status, the introduction of a stabilized GIP agonist and its potential impact on WAT metabolic activity in vivo could be indirect and involve hormone cascade pathways. Only a few of these molecules have been confirmed to play such a role[45] while gut-derived Fgf15, an important regulator of EE, can stimulate the BAT thermogenic program[46,47]. Our findings open up further prospects for exploiting the HNF4A-GIP regulatory loop for the treatment of metabolic disorders.

## Methods

**Animals.** Mice were maintained in standard housing conditions at 23 °C in a specific pathogen-free facility and received ad libitum food and water. Diets used are specified in the figure legends and included a regular chow (Charles River Rodent Diet 5075; 2.89 kcal/g) composed of 55.2% carbohydrates, 18% proteins, and 4.5% fat and a HFD (TestDiet 58V8; 4.65 kcal/g) composed of 41.2% carbohydrates, 21.3% proteins, and 23.6% fat. C57BL/6J-*Hnf4a*$^{loxP/+}$ mice[13] were first crossed with C57BL/6-*12.4KbVil*Cre mice[48] to generate *12.4KbVil*Cre/*Hnf4a*$^{loxP/+}$ mice, which were subsequently bred with *Hnf4a*$^{loxP/loxP}$ mice to produce *12.4KbVil*Cre/*Hnf4a*$^{loxP/loxP}$ (HNF4A$^{\Delta IEC}$) mutant mice and their controls. Mice were genotyped with DNA isolated from tail biopsies[12,49]. Experiments were carried out using 12–16-week-old mice kept with a limited number of mice per cage. Female mice were also used to study the effect of HFD on weight gain for both genotypes. For metabolic analyses, male mice were individually placed in metabolic cages and provided with food and water at libitum while being housed on a reverse

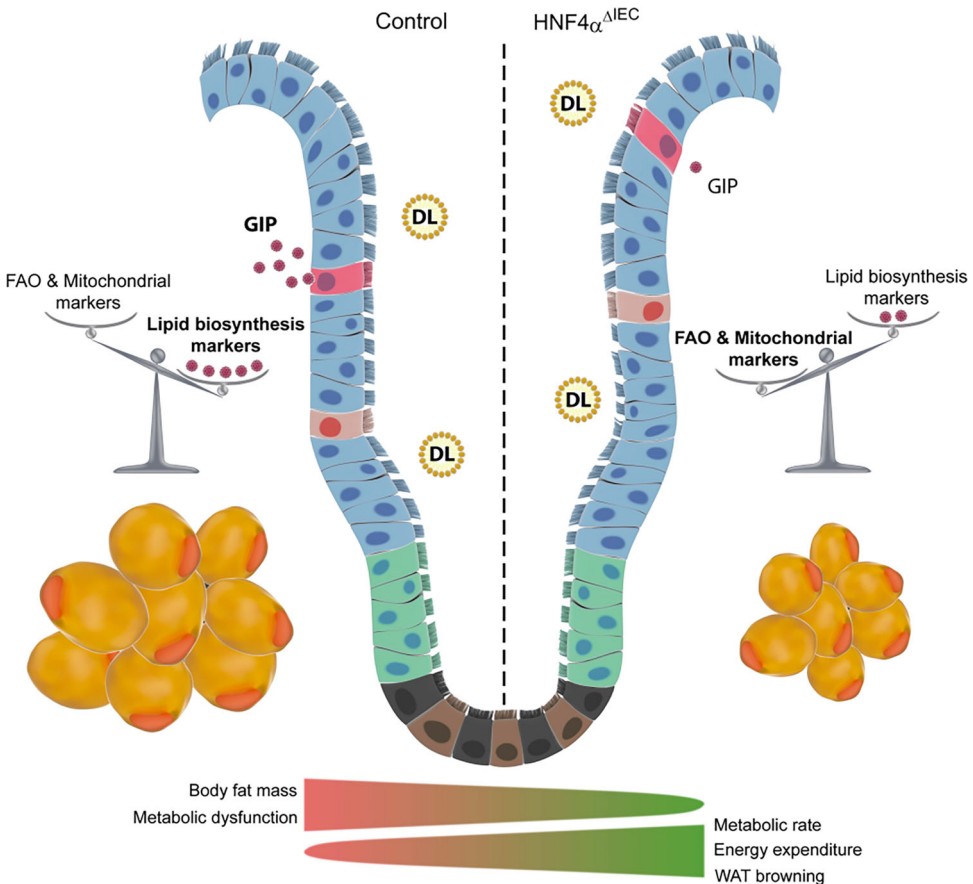

**Fig. 8 A model for gut-specific action of HNF4A on fat metabolism.** Schematic overview of the impact of HNF4A intestinal deletion on GIP production, energy expenditure, and WAT metabolism in mice fed an HFD.

light-dark cycle[13]. Plasma samples, organs, and tissues were stored at −80 °C prior to use. All procedures were approved by the Institutional Animal Research Review Committee of the Université de Sherbrooke (approval 102-18) and are compliant with animal Ethical Regulations. Body composition of post-mortem mice was measured using dual-energy X-ray absorptiometry with a PIXIMUS mouse densitometry apparatus (Lunar Corporation, Madison, WI, USA).

**Insulin and oral fat tolerance testing**. Mice fed a HFD were fasted for 4 h for fasting blood glucose measurements, or before an ITT or OFTT. ITT was performed subcutaneously at a 0.75 mUi/g body weight dose. Tail vein blood glucose was recorded prior to the assay and every 15 min for 2 h with a glucometer. OFTT was performed with olive oil at a dose of 5 μL/g body weight. Blood was sampled prior to oil gavage and after 15 and 30 min. Plasma was recovered by adding a 10% (vol/vol) anticoagulant mix to blood composed of 0.02 mg/μL EDTA (Sigma-Aldrich), 20% (vol/vol) DPP4 (Millipore), 10% (vol/vol) protease inhibitor cocktail (P8340, Sigma-Aldrich), 5 KIU/μL Aprotinin (AD0153, Bio Basic), and 0.2 M NaCl followed by immediate centrifugation at 3000 × g, 4 °C for 10 min. Total GIP (EMD Millipore, EZRMGIP-55K), GLP-1 (Crystal Chem, 81508), and resistin (ab205574) were measured using ELISA from plasma samples according to the manufacturer's instructions.

**Indirect calorimetry**. The Promethion High-Definition Room Calorimetry System was used for indirect calorimetry studies performed at 23 °C (GA3, Sable Systems. Las Vegas, NV). Prior to data collection, all animals were acclimated to cages for 3 days, followed by 4 days of data acquisition with a normal-fat diet to record the basal metabolic parameters. The mice were then exposed to HFD. To account for metabolic adaptation and stabilization to diet change, only the 3 last days of a 2-week HFD treatment were considered for data analysis. A standard 12-h light/dark cycle (6:00–18:00) was maintained throughout the calorimetry studies. Data acquisition and instrument control were coordinated using MetaScreen v. 2.3.0 Raw data were processed using ExpeData v. 1.8.4 (Sable Systems, Las Vegas, NV) and an analysis script that performs most of the aspects of data transformation. Lipid oxidation rates were calculated separately as previously described[13].

**PET-CT**. Mice were not fed for 5 h before PET/CT. All experiments were initiated immediately after insertion of a cannula into the tail vein for injection of PET tracers. All imaging was performed on an avalanche photodiode-based small animal PET scanner (LabPET/Triumph, Gamma Medica, Northridge, CA) at the Sherbrooke Molecular Imaging Centre. Mice were anesthetized with isoflurane (2.0%, 1.5 L min⁻¹). A bolus of [18F]-FDG (10 MBq, in 0.1 mL of 0.9% NaCl) was injected via the caudal vein over 60 s after starting a 20-min dynamic PET data acquisition, while the animal was maintained at 34 °C with a heated bed. Low-dose CT scan imaging was performed using the integrated X-O small-animal CT scanner of the Triumph platform, consisting of a 40 W X-ray tube with a 75 μm focal spot diameter and a 2240 × 2368 CsI flat panel X-ray detector. The detector pixel size was 50 μm and a 2 × 2-pixel binning scheme was used. Scans were performed at 60 kVp and 230 μA using 512 projections in the fly mode to reduce exposure. A dynamic series of 27 frames (12 × 10 s, 8 × 30 s, 6 × 90 s, and 1 × 300 s) was used for [18F]-FDG imaging. 3D images were reconstructed using 20 iterations of a maximum-likelihood expectation-maximization algorithm incorporating a physical description of the detector response functions. Regions of interest were drawn on short-axis images using Amide software v.1.0.5 and confirmed with the μCT scan. The glucose extraction coefficient (Ki) was determined using Patlak graphical analysis. These values represent the proportion of the circulating substrate taken up per tissue mass.

**(D-Ala²)GIP[Lys³⁷PAL]**. (D-Ala²)GIP[Lys³⁷PAL][29] was synthesized by GL Biochem Ltd. (Shanghai, China). Peptide integrity was confirmed upon reception using ultra-performance liquid chromatography-mass spectrometry. It was diluted in water for cell culture assays or in saline for injection into mice. Prior to diet change, mice were acclimated to three subcutaneous (s.c.) injections, distributed every 3 days, for 10 days on a regular chow diet. For mice exposed to the GIP analog, injections consisted of two pre-injections of saline followed by one sensitizer dose of (D-Ala²)GIP[Lys³⁷PAL]. HFD was then introduced for 2 weeks, during which mice received a total of 5 s.c. injections of saline or (D-Ala²)GIP[Lys³⁷PAL] at 2 pmol/kg. This dose was estimated to be in the physiological range of fasting circulating GIP levels in control mice (30 pM).

**Histology**. Liver and eWAT tissues were fixed in 4% paraformaldehyde overnight at 4 °C, dehydrated, embedded in paraffin, and cut into 5-μm sections. Jejunum samples were embedded in Tissue-Tek OCT compound, snap-frozen, and stored at −80 °C. The embedded samples were sectioned at 5 μm and stained with ORO.

Jejunum and eWAT slides were counterstained with hematoxylin and eosin. For ADFP and lipid-droplet staining, liver sections were immersed and boiled for 10 min in 10 mM citrate buffer (pH 6.0), washed, blocked with 2% BSA (Sigma-Aldrich) in PBS-Tween 0.1%, and incubated overnight at 4 °C with anti-ADFP (1:500, Ab52356). Slides were stained with anti-rabbit Alexa 488 conjugated secondary antibody (1:500, CST4412) and DAPI. Stained sections were analyzed using a NanoZoomer 2.0-RS (Hamamatsu Photonics, Japan) digital slide scanner, the LX2000 fluorescence module (Hamamatsu Photonics, Japan), and NDP.view software. Jejunum and adipose sections were visualized under visible light and adipocyte histomorphology was analyzed using CellProfiler (3.1.5).

**Real-time PCR**. Total RNA (RNeasy, Qiagen) from the jejunum, eWAT, and BAT was extracted according to the manufacturer's instructions, then cDNA was synthesized from 500 ng of total RNA using oligo(dT) primers and SuperScript IV reverse transcriptase (Invitrogen). The cDNA was amplified using real-time PCR (LightCycler, Roche) and data were normalized to the most stable reference genes assessed (https://www.heartcure.com.au/reffinder/): *Tbp* for jejunum; *Gapdh*, *Sdha*, and *Ubc* for BAT; *Ppia*, *Sdha*, and *Atpaf1* for eWAT. Transcripts were quantified relative to the control group value, which was set to 1 for normalization of several reference genes[50]. The eWAT gene expression panel is presented in the form of a heat map[51].

**Microarrays**. Probes were generated from total RNA isolated from jejuna from three control and three HNF4A$^{\Delta IEC}$ mutant mice, hybridized to an Affymetrix GeneChip Mouse Genome 430 2.0 Array using the microarray platform of McGill University and Génome Québec Innovation Center[49] (data are accessible through GEO series accession number GSE147105). Raw cel files were analyzed using Transcriptome Analysis Console software and tested for significant changes in signal intensity (eBayes ANOVA method followed by Benjamini–Hochberg step-up FDR-controlling procedure <0.10). GSEA was performed on the entire dataset using gene sets from Gene Ontology for lipid homeostasis (GO:0055088), lipid storage (GO:0019915), and KEGG for fat digestion and absorption (map04975)[52,53].

**RNA-seq**. Total RNA was extracted from cells using TRIzol (Invitrogen), following the manufacturer's protocol. Adipose tissues were collected in 1 mL of TRIzol. After the addition of chloroform, the aqueous layer was recovered, mixed with one volume of 70% ethanol, and applied directly to an RNeasy Mini Kit column (Qiagen). DNase treatment and total RNA recovery were performed according to the manufacturer's protocol. RNA integrity was assessed using an Agilent 2100 Bioanalyzer (Agilent Technologies). PolyA mRNA libraries were prepared from 200 ng of total RNA using an NEB E6420 kit, according to the manufacturer's recommendations. Library quality was evaluated on a Bionalyzer DNA HS Chip (Agilent) and quantified using a Qubit fluorimeter (Invitrogen). The libraries were then pooled and sequenced on an Illumina Nextseq500, PE40, at the RNomic Platform of Université de Sherbrooke. Quality control, quality, and adapter trimming of RNA-seq libraries were performed using Trim Galore version 0.6.4_dev (https://www.bioinformatics.babraham.ac.uk/projects/trim_galore/). Reads were aligned with Rsubread version 2.6.2[54] against mouse genome version GRCm38 in pair-end mode. Low-count genes were filtered with DEseq2 version 1.32.0[55] and differential expression analysis was performed using edgeR version 3.34.0[56]. Genes were retained when their FDR was less than 0.05 and absolute log fold change was greater than 1.5. They were then passed to DAVID version 6.8[57] to search for enriched GO, which also resulted in a list of GO that passed an FDR threshold of 0.05.

**Jejunum and fecal fat extraction**. Fresh fecal pellets and jejunum biopsies were processed for fat extraction using chloroform-methanol. Briefly, 50 mg of stool or tissue was disrupted in 1.2 mL of chloroform:methanol (2:1) and debris was pelleted for 5 min at 5000 × *g*). The supernatant was mixed with 300 μL of 0.9% NaCl. Following a quick centrifugation (20,000 × *g*, 30 s), the aqueous phase was discarded and the apolar phase was evaporated using a vacuum (SpeedVac Thermo Scientific). The remaining organic content was weighed and considered to be the total amount of fat recovered from the jejunal biopsies. Lipophilic matter from fecal samples was resuspended in 100 μL 1× NP40 reagent and TG amounts were quantified (10010303, Cayman Chemical) according to the manufacturer's instructions.

**Cell culture and differentiation**. 3T3-L1 preadipocytes were seeded at 5000 cells/well in a Seahorse XF96 microplate and cultured in 5% CO$_2$ at 37 °C in DMEM (4.5 g/L glucose; Wisent) supplemented with 10% FBS, 1% GlutaMax, 1% HEPES, and 1% penicillin-streptomycin. Cells were induced to differentiate 2 days after reaching confluence (day 0), by supplementing growth medium with 20 nM insulin (Humulin N; Eli Lilly), 0.25 μM dexamethasone (D4902, Sigma-Aldrich), and 2 μM rosiglitazone (R2408, Sigma-Aldrich) for 2 days with (10–500 pM) or without (D-Ala$^2$)GIP[Lys$^{37}$PAL] as indicated in the figure legends. This concentration range was chosen to reflect circulating GIP levels in response to OFTT of HNF4α$^{\Delta IEC}$ mutant (10 pM) and control (500 pM) mice. From days 2 through 8, the growth

medium was changed every 2 days and supplemented with only 20 nM insulin and the initial concentration of (D-Ala$^2$)GIP[Lys$^{37}$PAL].

**Seahorse analysis**. On day 8, the culture medium of mature 3T3-L1 adipocytes was replaced with Seahorse XF DMEM medium (103575-100) supplemented with 1 mM sodium pyruvate (Wisent), 2 mM glutamine (Wisent), 4.5 g/L glucose (Wisent), 20 nM insulin, and (D-Ala$^2$)GIP[Lys$^{37}$PAL], then incubated at 37 °C in a non-CO$_2$ incubator for 60 min. After hydration, the Seahorse cartridge was loaded with 1 μM oligomycin (port A), 1 μM carbonyl cyanide-4 (trifluoromethoxy) phenylhydrazone (FCCP; port B), 4 μM etomoxir (11969, Cayman) in DMSO 0.1% (port C), 1 μM rotenone (ab143145), and 1 μM antimycin (A8674, Sigma-Aldrich) in DMSO 0.001% and ethanol 0.002% (R&A; port D). OCR and ECARs were measured using an XF96 extracellular flux analyzer (Seahorse Bioscience). Cellular bioenergetics and parameter calculations were performed according to the manufacturer's instructions, with the etomoxir effect (OCR dedicated to FAO) calculated as maxOCRFCCP-meanOCRetomoxir.

**Statistics and reproducibility**. Unless otherwise specified, statistical analyses were performed and graphics were produced using GraphPad Prism 8 software. The data sets were compared using analysis of variance or non-parametric analysis when applicable. All statistical analyses are presented as *n* and mean ± SEM. No statistical method was used to predetermine sample size. No data were excluded from the analyses. Each experimental group was randomized during treatments. The Investigators were blinded to allocation during experiments and outcome assessment.

**Reporting summary**. Further information on research design is available in the Nature Research Reporting Summary linked to this article.

## Data availability
Microarray data are accessible through GEO series accession number GSE147105. RNA-seq data are accessible through GEO series accession number GSE189439. The website for GSEA is available on https://www.gsea-msigdb.org/gsea/index.jsp. Source data are provided with this paper.

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

## Acknowledgements
The authors thank Dr. Christine Lawson and Dr. Lee-Hwa Tai for providing access to the Seahorse XFe96 Analyzer, the Electron Microscopy & Histology Research Core of the FMSS at the Université de Sherbrooke for histology and phenotyping services, and Dr. Mariano Avino from the Bioinformatics platform of the Department of Biochemistry and Functional Genomics at the Université de Sherbrooke for bioinformatic services and the FRQS-funded « Centre de Recherche du CHUS» (CR-CHUS) for providing access to the indirect calorimetry platform. This research was funded by Canadian Institutes of Health Research (CIHR) grant PJT-156180 to F.B., Natural Sciences and Engineering Research Council of Canada (NSERC) grant RGPIN-2017-06096 to F.B., and a PAFI grant from CR-CHUS. R.G. is the recipient of an FRQS scholarship. A.C.C. is the recipient of the Canada Research Chair in Molecular Imaging of Diabetes.

## Author contributions
R.G., S.T., F.M.R., N.P., C.N., A.C.C., and F.B. conceived and designed the research; R.G., S.T., F.M.R., C.N., S.S.J., and Y.G. performed the experiments; R.G., S.T., C.N., C.J., M.L., A.C.C., and F.B. analyzed data; R.G., S.T., C.J., M.L., A.C.C., and F.B. interpreted results of the experiments; R.G., S.T., and C.J. prepared figures; R.G., C.N., A.C.C., and F.B. drafted the manuscript; R.G., C.N., M.L., A.C.C., and F.B. edited and revised the manuscript; R.G., S.T., F.M.R., C.N., S.S.J., C.J., Y.G., N.P., M.L., A.C.C., and F.B. approved the final version of the manuscript.

## Competing interests
The authors declare no competing interests.
