## [Peer Review File · Nature Communications]

The transcription factor Hepatocyte Nuclear Factor 4A acts in the intestine to promote white adipose tissue energy storageREVIEWER COMMENTS

Reviewer #1 (Remarks to the Author):

In this study, Girard et.al., examine how intestinal HNF4A maintains white adipose tissue and promotes diet-induced obesity in mouse models. They show conditional HNF4A knockout in the intestine prevents the development of HFD-induced obesity, despite having the same intake of fats as control mice with regular HNF4A function. They attribute this to browning of white adipose tissue under HFD in these mutants, and a preference for fat as an energy source, which ultimately causes less fat storage and increased energy expenditure. Gip mRNA and circulating GIP in the blood are decreased in the intestinal HNF4A mutant. The reduced DIO phenotype in the HNF4A mutant is rescued by GIP treatment, which results in weight gain and body fat composition comparable to control mice.

I find the work interesting, important in the context of obesity research, and it utilizes a nice blend of mouse genetics and genomics and physiology techniques to arrive at the conclusions. There are several substantial weaknesses to address before publication:

Comments:

1) Were controls set up with control and HNF4A KO mice under normal diets?

2) Why were only male mice used? Is this effect also seen in female mice? There have been reports of female mice being protected from HFD induced metabolic syndromes (PMID: 23049932). Can it be said that HNF4A has a role to play there?

3) Would it have been better to use inguinal WAT for the experiments in both genders so as to eliminate or highlight sex-specific differences?

4) In Fig 1(a), significant weight differences between control and KO under HFD are seen only at 8 weeks, whereas in Fig 2(e), the authors show the resistance to weight gain starts at 4 days. These results seem contradictory and could cause confusion. Are the bodyweights on control diet any different between WT and mutants? Rather than plot the ratio only, it would be helpful to plot the individual body weights (including mice on control diets) that lead to the analysis in 2E, maybe in a supplemental figure.

5) The authors say the GSEA of the 57 differentially expressed genes do not represent lipid homeostasis, fat digestion and absorption or lipid storage. So, which ontologies are highly represented? Some more information and follow-up about this would be nice. Why was GIP prioritized from this list?

6) In Fig S1(a), there seems to be a modest elevation of Ucp1 across the sample replicates. Can the authors provide quantitative measurements and statistics for the claim that thermogenesis transcripts are elevated in eWAT?

7) In Fig S1(b), AMPK levels do not seem to be much higher in the KO than the control. Also, the levels of the loading control, Vinculin, look to be higher in the KO, which could explain why AMPK is higher. I think the western blot needs to be repeated to clarify this result.

8) In fig 3(a), why was inguinal WAT FDG uptake similar between control and KO, while eWAT is higher in KO? I think this should be explained better.

9) What sort of adipocytes are 3T3-L1 cells? Are they representative of WAT? Can experiments be done to translate the Seahorse results seen in the cell lines to the control and HNF4A KO mice?

10) Are other cell lines or models available to corroborate the results? Can primary tissues from mouse be used?

11) The media used for 3T3-L1 cell culture is 4.5g/L high glucose media which equates to 25mM of glucose – almost 5 times normal physiological levels. Was this metabolically accounted for during experiments?

12) In Fig 4(e), it's hard to see any differences in proton leak. Can the authors add the bar graphs for it?

13) I feel like the dosages of GIP used need to be revised. Upon administration of HFD, the authors used 2pmol/kg to reach the physiological fasting level of 30mM. But in cell lines, they state that upon OFTT, the physiological levels are 500mM in control mice and they use this concentration. So, shouldn't the dose they used for mice under HFD be higher? This is confusing.

14) Upon GIP administration what happens to the adipose tissue? Is the browning stopped and is WAT renewed? This is not addressed. Is there an increase in mitochondrial content?

15) Generally speaking, bar graphs should have individual data points plotted, for best practices in scientific communication and clarity.

16) The Villin Cre line used should be indicated more clearly in the figures, and the body weights of control mice and mice on control diets should be reported to help us understand any baseline differences between the genotypes and conditions.

17) GIP levels decrease in HNF4A KO in mice on HFD. Does this also occur in mice on normal chow? The authors could also try and explain the mechanism through which HNF4A controls GIP levels – could this be direct or indirect? Cell autonomous or non-cell autonomous?

Reviewer #2 (Remarks to the Author):

The authors previously created a mouse with an intestinal deletion of HNF4A and showed that it had an intestinal epithelial phenotype and influenced GIP secretion (refs 7 & 8). HNF4A was also thought to be essential in several aspects of metabolism including lipid metabolism. Mutations are of course underlying the MODY 1 syndrome. They here report that is resistance to DIO and decreasing whole body energy expenditure (one would suspect a causal relationship here?) favoring fat combustion and perhaps changes in WAT browning. The main discovery of the study is that administration of a long-acting GIP analog rescued the phenotype of the mutant mouse tying together the previous observations of the abnormalities of GIP secretion with this metabolic phenotype.

Currently there is a big confusion in the literature about GIP, with one camp claiming, mainly based on rodent studies, that GIP agonists cause weight loss and reduce food intake (Mroz et al Mol Metab 2019) while another group (Killion et al endocrine review. 2020) thinks that GIP antagonism if anything would lead to weight loss (see also very recent paper by Samms R , TEM 2020). Of course, the authors cannot be blamed for this confusion, but it makes it very difficult to translate the current findings into something sensible. On top of this, the GIP analog used in the present studies was reported by the people who developed it not to have an influence of lipid metabolism. This may or may not be true, but makes the choice of this GIP analog debatable. The authors are right, of course, that a long-acting GIP analog must be used, but judging from the ones developed by Mroz et al, such analogs would promote fat loss rather than the opposite. In conclusion, I am not sure what the present study really shows. It seems convincing that the mouse model has an unusual phenotype but the association to GIP is not convincing. Other wishes the studies seem to have been conducted with care and the manuscript is well written. Specific points arising from going through the manuscript (unfortunately there are no line numbers)

In the abstract HNF4A is described as a nuclear receptor but in the start of the main part as a transcription factor. It might be useful for the readers if the authors would care to discuss whether there is a difference between the two designations.

p. 4. "Fecal rejection" of triglyceride is to me an unclear description.

Perhaps it should be mentioned that the epididymal fat of rodents is considered to belong to the visceral fat.

Is the term "non-cell autonomous mechanism" completely clear? I think the readers could use a bit of help here. And why is that so interesting?

I am questioning the dose of the GIP analog and the resulting exposure after this dose. How much? How long? To me the dose seems very small and it is important to know the duration of the exposure. (rodents break down GIP extremely rapidly, and although probably surviving longer in the body because of the acylation, does the analog remain bioactive? In addition, as already mentioned, this specific agonist was selected to show only effects on insulin secretion and not on fat metabolism.

It is stated that weight development was similar in the two groups of mice (deletions and controls) during treatment whereas the mutants showed less weight gain after saline, and fat development was also similar, but less in saline treated controls. This would confirm the adipogenic effects of the GIP analog but this is not what the group who developed the analog found.

P, 8 is FCCP defined? Could be described.

I am not sure that the effects of the GIP analog on the differentiated 3T3 preadipocytes is helpful at all.

And the conclusion that "GIP negatively impacts metabolic rates of adipocytes resulting in a reduction of energy expenditure and increase of fat energy storage in the WAT of mice fed and HFD" is not strongly supported by these data.

Further, the conclusion that this is consistent with the leanness of MODY 1 patients (as opposed to T2DM) is to compare pears and apples – the type 2DM patients get their diabetes because of their obesity, whereas the others primarily have a problem with insulin secretion .

The reference for MODY 1 patients and GIP secretion is misquoted – the patients did not have lower prandial GIP secretion compared to controls (but appeared to depend strongly on GIP for their insulin secretion) .

Point-by-point responses to reviewers

We thank the reviewers for their constructive comments as well as the editor for offering time flexibility during these challenging pandemic times. We are providing a revised version of the manuscript formatted to include separate sections for introduction, results, and discussion as well as additional experimental work to address the concerns raised by both reviewers. The original comments of the reviewers are in *italics*, while revised portions of the manuscript are indicated in **red**.

Reviewer #1:

In this study, Girard et.al., examine how intestinal HNF4A maintains white adipose tissue and promotes diet-induced obesity in mouse models. They show conditional HNF4A knockout in the intestine prevents the development of HFD-induced obesity, despite having the same intake of fats as control mice with regular HNF4A function. They attribute this to browning of white adipose tissue under HFD in these mutants, and a preference for fat as an energy source, which ultimately causes less fat storage and increased energy expenditure. Gip mRNA and circulating GIP in the blood are decreased in the intestinal HNF4A mutant. The reduced DIO phenotype in the HNF4A mutant is rescued by GIP treatment, which results in weight gain and body fat composition comparable to control mice.

I find the work interesting, important in the context of obesity research, and it utilizes a nice blend of mouse genetics and genomics and physiology techniques to arrive at the conclusions. There are several substantial weakness to address before publication:

We thank this reviewer for acknowledging the interest and importance of our findings in the field. We are providing revisions or clarification for each of the 17 following points that were raised.

1) *Were controls set up with control and HNF4A KO mice under normal diets?*

- We did not include controls because we have recently reported that these mice display similar nutritional behavior as well as energy metabolism (weight mass, adipose density, fat mass, etc.) on normal diets (Girard R. et al, 2019; DOI: [10.1038/s41598-019-41061-z](https://doi.org/10.1038/s41598-019-41061-z)). We now cite these data in the revised Introduction:

Page 3, lines 74-76

To investigate the intestinal role of HNF4A in this context, we used a murine conditional deletion (HNF4A^{ΔIEC}) previously shown to exhibit minor phenotypes without impacting nutrition, whole-body energy metabolism, and weight management under a normal diet^{12, 13}.

2) *Why were only male mice used? Is this effect also seen in female mice? There have been reports of female mice being protected from HFD induced metabolic syndromes (PMID: 23049932). Can it be said that HNF4A has a role to play there?*

Female HNF4A KO mice were also resistant to weight gain under HFD relative to female control mice. We have provided these data in a new **Fig. S1**. We performed preliminary indirect calorimetry with female mice, but they showed high variability between individuals, even within the control groups. This was not unexpected because more pronounced effects on several HFD-associated features were observed in males with the C57BL/6N phenotype (Ingvorsen C. Nutr Diabetes 2017 (DOI: 10.1038/nutd.2017.6);

Mauvais-Jarvis F. Cell Metabolism 2017 (DOI: 10.1016/j.cmet.2017.04.033). Although it will be important to consider sex issues in the future, we felt that addressing sex affects the resistance to weight gain in the absence of HNF4A via similar mechanisms would require a much larger number of experiments for each piece of data, keeping in mind that we would have to consider the inclusion of a larger number of additional female mice to account for significance as opposed to the males. Sex differences are definitely an important issue with regard to metabolism, and we have included the following perspective sentence in the discussion section to emphasize this aspect.

Page 13, lines 330-334

One important limitation of our study was its focus on male subjects. Although we observed that female HNF4A^{ΔIEC} mice displayed similar resistance to DIO, this observation does not warrant the conclusion that the same mechanism operates in both sexes, as recently noted⁴⁴. This aspect deserves further consideration, especially in the design of clinical studies to follow up on our findings.

3) *Would it have been better to use inguinal WAT for the experiments in both genders so as to eliminate or highlight sex-specific differences?*

As we noted above, the effect of sex in our studies was not investigated in detail, given the high variability in the indirect calorimetry data collected from females. We have included additional experiments, including both eWAT and iWAT (RNA-seq), based on specific changes observed in PET/CT imaging data that were observed in both groups of males (see points 10 and 7).

4) *In Fig 1(a), significant weight differences between control and KO under HFD are seen only at 8 weeks, whereas in Fig 2(e), the authors show the resistance to weight gain starts at 4 days. These results seem contradictory and could cause confusion. Are the bodyweights on control diet any different between WTs and mutants? Rather than plot the ratio only, it would be helpful to plot the individual body weights (including mice on control diets) that lead to the analysis in 2E, maybe in a supplemental figure.*

As requested, we are now providing a plot of individual body weights in a supplemental figure (Fig. S3). The ratios in Fig. 2E showed significance starting on day 4 in adults, but no significance was observed when individual body weights were analyzed. Due to a shorter HFD exposition in this experiment when compared to Fig 1a, we chose to represent weight gains rather than body weight to best exemplify the effect of HFD in these conditions.

We have clarified this observation in the revision:

Page 5, lines 121-122

HNF4A^{ΔIEC} mutants were significantly more resistant to weight gain as early as 4 days after commencement of HFD when ratios of the weights were plotted (Fig. 2e) as opposed to individual body weights (Fig. S3).

5) *The authors say the GSEA of the 57 differentially expressed genes do not represent lipid homeostasis, fat digestion and absorption or lipid storage. So, which ontologies are highly represented? Some more information and follow-up about this would be nice. Why was GIP prioritized from this list?*

GSEA did not identify any significant ontology, probably because of the small number of predicted altered genes. Thus, we performed a GO analysis on these 57 genes using the PANTHER webtool (<http://www.pantherdb.org>). Very few ontologies were identified as significant. The most significant GO was “regulation of peptide transport,” with nine molecules being modulated out of 243 (FDR < 1.47E-05). We have provided the details for these molecules in a new supplemental figure (Fig. S2) and added the following sentences to the Results section:

Page 4, lines 101-103

Gene ontology analysis identified very few biological processes, with regulation of peptide transport as the most significant functional annotation (FDR < 1.5 x 10⁻⁰⁵) (Fig. S2a).

Page 8, line 199

Gene ontology identified multiple potential HNF4A targets (Fig. S2b).

6) *In Fig S1(a), there seems to be a modest elevation of Ucp1 across the sample replicates. Can the authors provide quantitative measurements and statistics for the claim that thermogenesis transcripts are elevated in eWAT?*

This supplemental figure has been integrated into the manuscript and is now a part of Fig. 3. The quantification is now provided in Fig. 3b, where only Ucp1 and Prdm16 showed significance. We have modified the text accordingly:

Page 7, lines 156-157

Ucp1 and Prdm16 were induced 14.5-fold (P < 0.05) and 1.5-fold (P < 0.01), respectively (Fig. 3b).

7) *In Fig S1(b), AMPK levels do not seem to be much higher in the KO than the control. Also, the levels of the loading control, Vinculin, look to be higher in the KO, which could explain why AMPK is higher. I think the western blot needs to be repeated to clarify this result.*

We agree and thank you for pointing out this discrepancy. We have performed additional western blots using multiple total-protein extracts from eWAT. Although we observed a significant increase in AMPK phosphorylation in KO eWAT (2.7-fold; *P* < 0.05), the overall quality of each western blot was not satisfactory. Protein stability appears to be an issue. For this reason, we feel more secure in removing both the AMPK and PPARα data that had been in Fig. S1.

8) *In fig 3(a), why was inguinal WAT FDG uptake similar between control and KO, while eWAT is higher in KO? I think this should be explained better.*

These observations indicate that only eWAT glucose uptake is modified in HNF4A mutants under HFD. However, when compared, FDG avidity showed that both iWAT and eWAT behaved similarly to BAT in the absence of HNF4A. This is intriguing, but logical, given the hypothesis that WAT becomes more metabolically active in the absence of intestinal HNF4A under HFD. An interesting link with GIP signaling has been reported that could explain this observation. This rationale is now better explained in the Discussion section:

Pages 12-13, lines 302-310

Our data demonstrate that in the absence of gut HNF4A, mice fed an HFD displayed an increase in overall WAT metabolic activities at comparable levels to BAT activity. In addition, the injection

of a GIP analog potentially influenced the WAT transcriptome with modifications associated with a reduction in fatty-acid metabolism, oxidative phosphorylation, and mitochondrial beta-oxidation. Taken together, these observations support the natural positive adiposity potential of GIP in WAT. In accordance with this, *Gipr* expression was reported to be more important in WAT than in BAT⁴⁰. Whether GIP signaling preferentially inhibits the metabolic activity of WAT based on the availability of GIPR is possible in this context.

9) *What sort of adipocytes are 3T3-L1 cells? Are they representative of WAT? Can experiments be done to translate the Seahorse results seen in the cell lines to the control and HNF4A KO mice?*

Several studies have used 3T3-L1s as a model to investigate the WAT. Interestingly, one of them has confirmed that these cells, when placed under the same differentiation protocol that we used, are mostly equivalent to eWAT, as validated by transcriptomics (Morrison and McGee Adipocyte 2015 (DOI: [10.1080/21623945.2015.1040612](https://doi.org/10.1080/21623945.2015.1040612))). We have clarified this point in the revised Results :

Page 9, lines 227-229

Murine 3T3-L1 adipocytes are widely used as a model of WAT and have been found to be typical of white adipocytes³¹.

It is not expected that performing experiments with isolated adipocytes from control and HNF4A KO mice would show any basal differences in the proposed experimental setup. The paracrine effect of GIP produced by the intestine, in which HNF4A deletion is restricted, would no longer be in place, regardless of the genotype source of isolated adipocytes. However, we performed an additional experiment to validate our findings (see next point).

10) *Are other cell lines or models available to corroborate the results? Can primary tissues from mouse be used?*

As explained above, 3T3-L1 cells are widely used as a cell line, mostly recapitulating white adipocytes. Attempts to culture primary cells from adipose tissue have been technically challenging. We have thus alternatively attempted to address this concern by investigating the effect of injecting a GIP analog into mice on both iWAT and eWAT transcriptomes. These data are now included in a new figure (Figure 7). The following text has been added to the Results section:

Pages 10-11, lines 250-259

To further investigate the effect of (D-Ala²)GIP[Lys³⁷PAL] on WAT metabolism, mice were exposed to the same protocol described above (Fig. 5c). A transcriptomic analysis was performed on both iWAT and eWAT from mice injected or not injected with the agonist. This analysis identified 562 genes for iWAT (Fig. 7a) and 75 genes for eWAT (Fig. 7b) that were differentially expressed (FDR < 0.05) between control and (D-Ala²)GIP[Lys³⁷PAL]-injected mice. GSEA showed a significant reduction in fatty-acid metabolism (Fig. 7c), oxidative phosphorylation (Fig. 7d), mitochondrial beta-oxidation (Fig. 7e), and adipogenesis (Fig. 7f) of iWAT after agonist treatment. Similar results were also observed in eWAT, with significant reductions in fatty-acid metabolism (Fig. 7g) and oxidative phosphorylation (Fig. 7h).

11) *The media used for 3T3-L1 cell culture is 4.5g/L high glucose media which equates to 25mM of glucose – almost 5 times normal physiological levels. Was this metabolically accounted for during experiments?*

This high concentration of glucose is required to induce and maintain 3T3-L1 adipocyte differentiation, and it is the standard in the literature. Thus, it is difficult to test other glucose concentrations without impacting the differentiated phenotype of these cells during experiments.

12) *In Fig 4(e), it's hard to see any differences in proton leak. Can the authors add the bar graphs for it?*

You are correct to note that proton leak (also designated uncoupled) is not noticeable in Fig. 4e. However, the former Fig. 4g (now Fig. 6c) shows bar graphs illustrating this difference.

13) *I feel like the dosages of GIP used need to be revised. Upon administration of HFD, the authors used 2pmol/kg to reach the physiological fasting level of 30mM. But in cell lines, they state that upon OFTT, the physiological levels are 500mM in control mice and they use this concentration. So, shouldn't the dose they used for mice under HFD be higher? This is confusing.*

The exact analog concentration that matches the physiological levels is not intuitive in this context. The analog is modified to increase its stability in circulation for several hours, while levels reached in a prandial timeframe are normally transient, given the short half-life of natural endogenous GIP. Thus, it is not an easy task to optimize the dose while comparing mutant and control mice exposed to HFD. Our rationale was to aim at a concentration near the basal levels to prevent supraphysiological levels in the mouse control population. One could argue that injecting too much of a stabilized GIP analog on top of naturally induced levels of endogenous GIP during a high-fat meal could interfere with the control condition in these experiments. In our opinion, it would be unwise to provide constant equivalent prandial doses in this *in vivo* setup and risking supraphysiological non-specific effects. The cell line experiments do not account for long-term storage effects (the experiment was performed for 100 min after a 60 min exposure to GIP agonist) in contrast to animals (24 days of GIP agonist injection).

14) *Upon GIP administration what happens to the adipose tissue? Is the browning stopped and is WAT renewed? This is not addressed. Is there an increase in mitochondrial content?*

RNA-seq indicated that GIP analog administration influenced both the eWAT and iWAT transcriptome after HFD exposure (Figure 7). GSEA showed a reduction in fatty acid metabolism, oxidative phosphorylation, and mitochondrial beta-oxidation following GIP analog administration. These data point again toward the positive adiposity potential of GIP in WAT.

15) *Generally speaking, bar graphs should have individual data points plotted, for best practices in scientific communication and clarity.*

This is a valid point. All bar graphs now include individual data points plotted as you suggest.

16) *The Villin Cre line used should be indicated more clearly in the figures, and the body weights of*

control mice and mice on control diets should be reported to help us understand any baseline differences between the genotypes and conditions.

The VillinCre/Hnf4a model (HNF4A^{ΔIEC}) is now properly defined in the new Introduction section to avoid confusion. We have chosen to keep the HNF4A^{ΔIEC} designation in all figures to be consistent with the literature, including our own previously published work.

Page 4, lines 77-79

This mouse model uses the *VillinCre* transgene to drive Cre expression exclusively in the intestinal epithelium since homozygosity for the unconditional null *Hnf4a* allele is embryonic lethal

We did not include body weights of control and mutant mice on control diets, since these groups did not show any significant changes in body weights, as we previously reported in two independent studies (Babeu JP et al., 2008 (DOI:); Girard R et al., 2019 (DOI:)). We felt that this was a repetition of the experiments already published by our group.

17) *GIP levels decrease in HNF4A KO in mice on HFD. Does this also occur in mice on normal chow? The authors could also try and explain the mechanism through which HNF4A controls GIP levels – could this be direct or indirect? Cell autonomous or non-cell autonomous?*

These are important points, and we apologize if the original letter format of the manuscript was not sufficiently clear to this end. We have now extended these inquiries in the revised **Discussion**:

Pages 11-12, lines 277-285

GIP is exclusively produced by the small intestine, and we recently showed that HNF4A acts as a positive regulator of *Gip* transcription¹³. We also found a modest reduction in circulating GIP levels after an oral glucose tolerance test, with no effect on glucose homeostasis in mice deficient for HNF4A in the intestine¹³. These observations are in contrast with our observation of GIP secreted levels being drastically reduced in HNF4A^{ΔIEC} mutant mice after lipid exposure. In addition to its transcriptional role on *Gip*, it is most likely that HNF4A plays a complementary role in GIP secretion after exposure to a lipid bolus; however, the exact nature of these mechanisms awaits further elucidation.

Reviewer #2

1) The authors previously created a mouse with an intestinal deletion of HNF4A and showed that it had an intestinal epithelial phenotype and influenced GIP secretion (refs 7 & 8). HNF4A was also thought to be essential in several aspects of metabolism including lipid metabolism. Mutations are of course underlying the MODY 1 syndrome. They here report that is resistance to DIO and decreasing whole body energy expenditure (one would suspect a causal relationship here?) favoring fat combustion and perhaps changes in WAT browning. The main discovery of the study is that administration of a long-acting GIP analog rescued the phenotype of the mutant mouse tying together the previous observations of the abnormalities of GIP secretion with this metabolic phenotype.

Currently there is a big confusion in the literature about GIP, with one camp claiming, mainly based on rodent studies, that GIP agonists cause weight loss and reduce food intake (Mroz et al Mol Metab 2019) while another group (Killion et al endocrine review. 2020) thinks that GIP antagonism if anything would lead to weight loss (see also very recent paper by Samms R, TEM 2020). Of course, the authors cannot be blamed for this confusion, but it makes it very difficult to translate the current findings into something sensible. On top of this, the GIP analog used in the present studies was reported by the people who developed it not to have an influence of lipid metabolism. This may or may not be true, but makes the choice of this GIP analog debatable. The authors are right, of course, that a long-acting GIP analog must be used, but judging from the ones developed by Mroz et al, such analogs would promote fat loss rather than the opposite. In conclusion, I am not sure what the present study really shows. It seems convincing that the mouse model has an unusual phenotype but the association to GIP is not convincing.

We agree completely with your description of this confusion and suggest that the nature of GIP agonists may provide an explanation. However, many rodent studies have suggested that inhibition of the GIPR pathway promotes weight loss, which is consistent with our data. These studies are cited in the Discussion. The Mroz et al. study is interesting, as they found that a new generation of GIP agonists can cause weight loss in already obese mice. More importantly, they found this effect to be dependent on food consumption. This suggests different mechanisms depending on metabolic context. The obesity resistance of our MODY1 intestinal-only mouse model was not associated with any change in food consumption (Fig. 2a). Our experimental design was to reintroduce nearly physiological levels of the GIP agonist to mice that were not initially obese, but rather during the first 2 weeks of HFD exposure. Conflicting published data, particularly in the field of metabolism, are common. As you point out, it would be difficult to blame us for this confusion. We have logically designed the progression of our experimental work to better understand our observations within the limit of our model, which recapitulates some aspects of human MODY1; notably the tendency to remain lean, unlike type 2 diabetes. In the revised manuscript, we have included additional data that demonstrate an increase in adiposity in WAT following injection of the GIP analog (new Figure 7). We have included the following text in the revised Discussion:

Page 13, lines 312-328

There is currently a great deal of controversy over the physiopathological impacts of pharmacologically targeting GIP in the context of weight management, as well as for its therapeutic potential during diabetes, metabolic syndrome, and obesity⁴¹. One study that used the same GIP agonist reported a lack of any effect on total body fat composition when administered to mice that were already obese²⁹. More recently, newly optimized GIP analogs were able to modestly reduce body weight gain in steady state obese mice; this effect was strictly dependent on food intake changes⁴². One important difference with our study is that our

mouse model is severely impaired in GIP secretion upon exposure to dietary fat. Other data consistent with our observations support the anti-obesity effects of GIPR antagonists, at least in mice⁴³. To add to this controversy, there is a great deal of divergence in the literature regarding the physiological versus supraphysiological doses for GIP agonists. In terms of animal physiology, the basal circulating level of GIP ranges between 10 and 40 pM and can reach 100–500 pM after lipid ingestion. The observed anti-diabetic effects of some GIP agonists have been observed at doses up to a thousand times higher than the normal physiological circulating levels for GIP. We administered a GIP analog at close to the normal physiological level and thus must be interpreted in this context.

2) *Other wishes the studies seem to have been conducted with care and the manuscript is well written. Specific points arising from going through the manuscript (unfortunately there are no line numbers). In the abstract HNF4A is described as a nuclear receptor but in the start of the main part as a transcription factor. It might be useful for the readers if the authors would care to discuss whether there is a difference between the two designations.*

We have replaced “nuclear receptor” with “transcription factor” in the abstract. We have also clarified how a nuclear receptor such as HNF4A functions as a transcription factor in the revised Introduction:

Page 3, lines 60-62

Nuclear receptors belong to a superfamily of ligand-dependent transcription factors that play crucial roles in metabolic disorders⁶. Hepatocyte nuclear factor 4A (HNF4A; also designated MODY1) is a transcription factor of the nuclear receptor family that is expressed in multiple epithelial tissues of the liver, gut, pancreas, and kidney.

3) *p. 4. “Fecal rejection” of triglyceride is to me an unclear description.*

We have modified this term to “Fecal triglyceride content” (Page 5, line 115).

4) *Perhaps it should be mentioned that the epididymal fat of rodents is considered to belong to the visceral fat.*

As you suggest, we now include the following phrase:

Page 5, lines 126-127

...which is considered to belong to visceral fat in rodents,...

5) *Is the term “non-cell autonomous mechanism” completely clear? I think the readers could use a bit of help here. And why is that so interesting?*

We have clarified this term in the revision:

Page 8, line 198

Our data support a role for intestinal HNF4A in WAT metabolism via non-cell-autonomous action involving a signaling molecule originally produced by the intestinal epithelium.

6) *I am questioning the dose of the GIP analog and the resulting exposure after this dose. How much? How long? to me the dose seems very small and it is important to know the duration of the exposure.*

(rodents break down GIP extremely rapidly, and although probably surviving longer in the body because of the acylation, does the analog remain bioactive? In addition, as already mentioned, this specific agonist was selected to show only effects on insulin secretion and not on fat metabolism.

The precise dose and duration of the GIP analog are provided in the Methods section. To make it clearer, a timeline is included in the new Figure 5C. Indeed, endogenous GIP is usually very unstable, but the acylated GIP analog has been reported to be stable for more than 24 h when exposed to DPP4 *in vitro* (Irwin et al., 2006; DOI: 10.1021/jm0509997). We tentatively tested the GIP stability using ELISA, but found that the modified analog was not detected by classical GIP antibodies. Whether this analog can influence fat metabolism remains a matter of debate. Martin et al. (2013; DOI: 10.1016/j.bbagen.2013.03.011) showed that this agonist did not affect body-fat percentage. However, this experiment was performed on wild-type mice that were obese after **140 days of a HFD**. HFD can increase GIP levels in normal mice (see Figure 5B); thus, it is possible that injecting “physiological levels” of GIP analog was not able to further impact WAT in their experimental system. Our experimental approach used a MODY1 gut-specific model that was mostly deficient in producing GIP after lipid exposure (see Figure 5B). Our model recapitulates what was found in the GIPR KO model exposed to HFD (Miyawaki et al., 2002). Lastly, we have provided additional evidence that the GIP analog changes the WAT transcriptome (new Figure 7) in the same experimental context, as detailed in Figure 5C during the early stages of HFD exposure. The new Discussion section addresses these concerns.

Pages 12-13, lines 302-328

Our data demonstrate that in the absence of gut HNF4A, mice fed a HFD displayed an increase in overall WAT metabolic activities at comparable levels to BAT activity. In addition, the injection of a GIP analog potentially influenced the WAT transcriptome with modifications associated with a reduction in fatty-acid metabolism, oxidative phosphorylation, and mitochondrial beta-oxidation. Taken together, these observations support the natural positive adiposity potential of GIP in WAT. In accordance with this, *Gipr* expression has been reported to be more important in WAT than in BAT⁴⁰. Whether GIP signaling preferentially inhibits the metabolic activity of WAT based on the availability of GIPR is possible in this context.

There is currently a great deal of controversy over the physiopathological impacts of pharmacologically targeting GIP in the context of weight management as well as for its therapeutic potential during diabetes, metabolic syndrome, and obesity⁴¹. One study that used the same GIP agonist reported a lack of any effect on total body fat composition when administered to mice that were already obese²⁹. More recently, newly optimized GIP analogs were able to modestly reduce body weight gain in steady state obese mice; this effect was strictly dependent on food intake changes⁴². One important difference with our study is that our mouse model is severely impaired in GIP secretion upon exposure to dietary fat. Other data consistent with our observations support the anti-obesity effects of GIPR antagonists, at least in mice⁴³. To add to this controversy, there is a great deal of divergence in the literature regarding the physiological versus supraphysiological dose usage for GIP agonists. In terms of animal physiology, the basal circulating level of GIP ranges between 10 and 40 pM and can reach 100–500 pM after lipid ingestion. The observed anti-diabetic effects of some GIP agonists have been observed at doses up to a thousand times higher than the normal physiological circulating levels

for GIP. We administered a GIP analog at close to the normal physiological level and thus our data must be interpreted in this context.

7) *It is stated that weight development was similar in the two groups of mice (deletions and controls) during treatment whereas the mutants showed less weight gain after saline, and fat development was also similar, but less in saline treated controls. This would confirm the adipogenic effects of the GIP analog but this is not what the group who developed the analog found.*

Martin et al. tested their compound on obese mice exposed to HFD for 140 days as described above. It is thus not surprising that in their hands, the analog failed to further stimulate adipogenesis in mice already producing high levels of GIP. Moreover, the dose used in their experiments outranged the normal endogenous GIP levels and may not be assimilated to the physiological effects of GIP. Nevertheless, the experimental fact that these mice did not show anti-adipogenic effects again suggests that GIP would not cause weight loss in this context (see point 1 of this reviewer). As explained in point 6 above, this is addressed in the revised Discussion.

8) *P, 8 is FCCP defined? Could be described.*

As suggested, we defined the abbreviation as follows:

Page 10, lines 238-239

Maximal mitochondrial respiration induced by **carbonyl cyanide-4 (trifluoromethoxy) phenylhydrazone** (FCCP)...

9) *I am not sure that the effects of the GIP analog on the differentiated 3t3 preadipocytes is helpful at all. And the conclusion that “GIP negatively impacts metabolic rates of adipocytes resulting in a reduction of energy expenditure and increase of fat energy storage in the WAT of mice fed and HFD” is not strongly supported by these data.*

Seahorse experiments on differentiated white adipocytes demonstrated the potential of the GIP analog to reduce maximal mitochondrial respiration as well as uncoupled respiration, both of which are directly linked to energy expenditure. However, our experiments performed on the cell line did not account for fat energy storage. With the inclusion of new data on the effect of the GIP analog on WAT in mice (new **Figure 7**), we have based this conclusion on additional results:

Page 11, lines 262-263

Taken together, our data point to a mechanism for the GIP analog negatively affecting the metabolic rate of white adipocytes, resulting in a reduction in energy expenditure and an increase in fat energy storage in the WAT of mice fed an HFD.

10) *Further, the conclusion that this is consistent with the leanness of MODY 1 patients (as opposed to T2DM) is to compare pears and apples – the type 2DM patients get their diabetes because of their obesity, whereas the others primary have a problem with insulin secretion.*

We have been reluctant to compare/contrast MODY1 with T2DM. The latter is a very complex set of diseases and we did not perform any experiments with the aim of supporting any type of conclusion or speculation to this end. We are focusing on MODY1, which is more relevant to our investigation.

11) *The reference for MODY 1 patients and GIP secretion is misquoted – the patients did not have lower prandial GIP secretion compared to controls (but appeared to depend strongly on GIP for their insulin secretion).*

You are correct; thank you for pointing out our error. As explained above, we have better explained the potential impact of our work in the context of MODY1 in the revised Discussion. We have removed this reference and now include two independent references to better support moving the field ahead by focusing on the potential role of GIP/incretins in the pathophysiology of MODY1/HNF4A.

Page 12, lines 289-297

Although currently, there has been no direct functional association reported between HNF4A and GIP expression in the context of these diseases, some case reports have suggested possible relationships between GIP altered circulating levels in MODY1 patients. One described the discovery of novel *RFX6* variants enriched in a MODY1 cohort that were associated with lower fasting and stimulated levels of GIP³⁷. Another described a young patient bearing a mutation in *HNF4A* (MODY1) with an impaired incretin response that includes GIP³⁸. Determining whether such a regulatory circuit between HNF4A and GIP is of functional importance in normal and diseased obese subjects requires further investigation.

REVIEWERS' COMMENTS

Reviewer #1 (Remarks to the Author):

The authors have substantially improved the manuscript, and while I don't have the expertise to weigh in on Reviewer 2's concern about use of the GIP analog, I find the research to be of high quality and rigor and of interest and importance to the field.

Reviewer #2 (Remarks to the Author):

No further comments

Point-by-point responses to reviewers

We thank the reviewers for their recommendation to publish our work as resubmitted with our responses to their specific points.

Reviewer #1:

The authors have substantially improved the manuscript, and while I don't have the expertise to weigh in on Reviewer 2's concern about use of the GIP analog, I find the research to be of high quality and rigor and of interest and importance to the field.

We thank this reviewer for the provided comments.

Reviewer #2

No further comments

We thank this reviewer for the provided comment.